# Diurnal cloud cycle biases in climate models

Jun Yin [1,2] & Amilcare Porporato [1,2]

Clouds' efficiency at reflecting solar radiation and trapping the terrestrial radiation is strongly modulated by the diurnal cycle of clouds (DCC). Much attention has been paid to mean cloud properties due to their critical role in climate projections; however, less research has been devoted to the DCC. Here we quantify the mean, amplitude, and phase of the DCC in climate models and compare them with satellite observations and reanalysis data. While the mean appears to be reliable, the amplitude and phase of the DCC show marked inconsistencies, inducing overestimation of radiation in most climate models. In some models, DCC appears slightly shifted over the ocean, likely as a result of tuning and fortuitously compensating the large DCC errors over the land. While this model tuning does not seem to invalidate climate projections because of the limited DCC response to global warming, it may potentially increase the uncertainty of climate predictions.

[1] Department of Civil and Environmental Engineering, Princeton University, Princeton, NJ 08544, USA. [2] Princeton Environmental Institute, Princeton University, Princeton, NJ 08544, USA. Correspondence and requests for materials should be addressed to A.P. (email: aporpora@princeton.edu)

As efficient modulators of the Earth's radiative budget, clouds play a crucial role in making our planet habitable[1]. Their response to the increase in anthropogenic emissions of greenhouse gases will also have a substantial effect on future climates, although it is highly uncertain whether this will contribute to intensify or alleviate the global warming threat[2]. Such uncertainties are well recognized in the state-of-the-art general circulation models (GCMs)[3] and are typically associated with their performance in simulating some critical cloud features, such as cloud structure and coverage[4]. Among these features, perhaps the most overlooked one is the diurnal cycle of clouds (DCC), describing how certain cloud properties (e.g., cloud coverage) change throughout the day at a given location. Due to the clouds' interference with the diurnal fluctuations of solar and terrestrial radiation, shifts in DCC have the potential to strongly affect the Earth's energy budget, even when on average the daily cloud

**Fig. 1** Diurnal cycle of cloud climatology and its indexes. **a** Examples of diurnal cycle of average cloud coverage near Guangde, Anhui, China (30.7N, 119.2E) in summer (June, July, and August) averaged over 1986–2005. The vertical dot lines and horizontal dash lines show the centroid and mean of the diurnal cycle climatology; the shaded blue areas indicate plus and minus one standard deviation. More examples are presented in Supplementary Fig. 2. Empirical probability density function (PDF) of **b** mean ($\mu$), **c** standard deviation ($\sigma$), and **d** centroid ($c$) of diurnal cycle of cloud coverage climatology over the land (black solid lines) and ocean (blue dash lines) in all four seasons over 60S–60N. The data sources (satellite observations, reanalysis, and climate models) are indicated on the left side of the figure (see Supplementary Table 1 for a list of these)

coverage is the same[5]. Although the diurnal cycle of atmospheric convection has recently attracted more research attention[6–12], DCC has yet to be analyzed at a global scale to fully understand its radiative effects on the Earth's energy budget.

In this work, to assess the degree with which climate models capture the key features of the DCC, we calculate three main statistics describing the typical DCC in each season in climate model outputs and compare them with those obtained from satellite observations and reanalysis data. We show how the resulting DCC model discrepancies influence the global radiation balance, contributing to increased uncertainties in climate projections.

## Results

**Errors of DCC.** We focus on the total cloud coverage (a brief discussion of the effects of other cloud properties can be found in the last section), whose diurnal cycle is closely related to that of total cloud water path[13, 14] and thus plays a critical role in the energy budget. To avoid dealing with higher harmonics of a Fourier decomposition of the DCC for cases with significant deviations from sinusoidal shapes[15], here we focus on the standard deviation ($\sigma$), centroid ($c$), and mean ($\mu$) to capture the amplitude, phase, and the daily average of cloud coverage (Fig. 1a and "Methods" section). Note, however, that the centroid and standard deviation are usually very similar to the amplitude and phase of the first harmonic (see a comparison map in Supplementary Fig. 1). These three indexes of cloud climatology are computed for the outputs of the GCMs participating in the Fifth Phase of the Coupled Model Intercomparison Project, and then are compared with those from the International Satellite Cloud Climatology Project (ISCCP)[16] and from the European Centre for Medium-Range Weather Forecasts (ECMWF) twentieth century reanalysis (ERA-20C)[17], all of which have high-frequency (3-h) global coverage for the period 1986–2005. ISCCP records were obtained from both visible and infrared channels; the latter is used to derive the cloud coverage as the infrared is measured throughout the whole diurnal cycle[18, 19]. While we are well aware that the ISCCP satellite records contain artifacts that may affect long-term trends[20, 21], it is important to emphasize that they do provide very useful information about the cloud climatology of interest here. In fact, it has been shown that the DCC climatology from these ISCCP records is generally consistent with the observations from stationary weatherships and some other satellite records[19, 22, 23]. Regarding ERA-20C, it is the ECMWF's state-of-the-art reanalysis designed for climate applications[17]. Both ERA-20C and CNRM-CM5 climate models rely on the integrated forecast system from ECMWF[17, 24], so that some common elements of cloud climatology may be expected.

Figure 1a shows an example of DCC climatology and the corresponding indexes for a subtropical monsoon climate zone in Eastern China in summer, characterized by clear mornings and frequent afternoon thunderstorms. This type of diurnal cycle is evident from the satellite (Fig. 1a ISCCP), reanalysis data (Fig. 1a ERA-20C), and CNRM-CM5 (Fig. 1a CNRM-CM5), while most of the GCMs show inconsistencies (see more examples in Supplementary Fig. 2). To explore these discrepancies globally, we calculated the DCC indexes at each grid point in each season from each data source. The most striking feature of DCC indexes is the land/ocean patterns, reflecting the contrasting mechanisms of atmospheric convection, although these geographical patterns are less coherent in the GCM outputs (see Supplementary Figs. 3–13). For this reason, we compare the empirical distributions of DCC indexes over the land and the ocean in Fig. 1. The satellite, reanalysis, and CNRM-CM5 clearly show larger $\mu$, smaller $\sigma$, and earlier $c$ over the ocean. A consistent pattern is found in all GCM

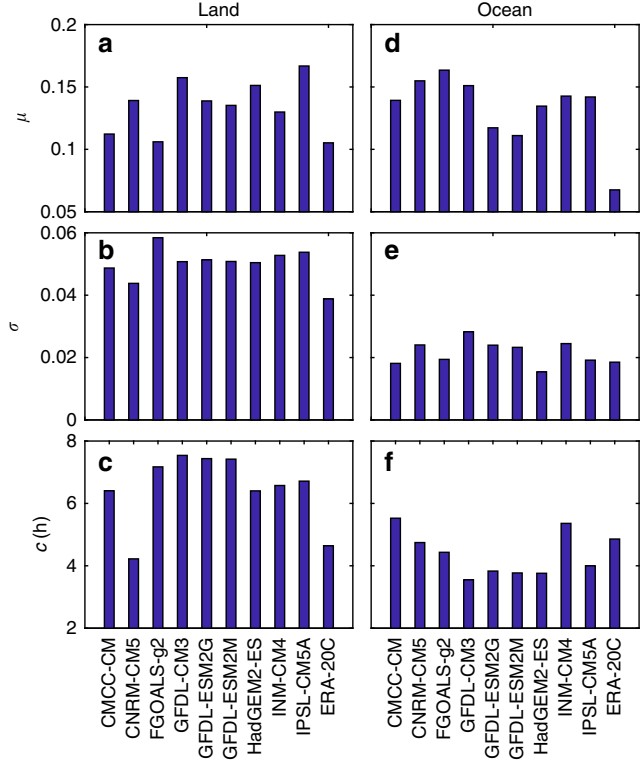

**Fig. 2** Root-mean-square deviation of DCC indexes between satellite observations and climate model outputs. **a**, **d** RMSD of mean cloud coverage, $\mu$, over the land and ocean; **b**, **e** RMSD of cloud coverage standard deviation, $\sigma$, over the land and ocean; **c**, **f** RMSD of the cloud coverage centroid, $c$, over the land and ocean

outputs for the mean cloud coverage (Fig. 1b). However, the DCC amplitude $\sigma$ generally shows no significant land–ocean contrast and a number of GCMs erroneously suggest stronger DCC amplitude over the oceans (Fig. 1c), while regarding the phase $c$, the land–ocean contrast is underestimated with most GCMs not even capturing the afternoon cloud peaks (Fig. 1d). Overall, only CNRM-CM5 shows reasonable simulation of DCC over the land, likely due to its detailed convective schemes and model resolution[24].

A detailed analysis of these differences is given in Fig. 2, which compares the root-mean-square deviation (RMSD, see "Methods" section) of $\mu$, $\sigma$, and $c$ in climate models and reanalysis data with the standard values from ISCCP records. Regarding the mean, $\mu$, the discrepancies over the land and ocean are relatively similar. The corresponding Taylor diagrams[25] further suggest that the mean cloud coverage is much better simulated than the rest DCC indexes (see Supplementary Figs. 14 and 15). As for $\sigma$ and $c$, the RMSDs between ISCCP records and the other data sets over the ocean are clearly smaller than those over the land. In these continental regions, the CNRM-CM5 model and the reanalysis data ERA-20C show relatively smaller RMSD. In general, models with obvious similarities in code produce a similar cloud climatology and RMSD (e.g., CNRM-CM5 and ERA-20C; GFDL-ESM2M and GFDL-ESM2G). A more detailed comparison between each data source is presented in Supplementary Fig. 16.

**Controls of cloud cycle on radiation budget.** Given the discrepancies in DCC predictions, it is logical to wonder how they may influence the Earth's radiation budget. To address this point, we follow and extend the approach of the so-called cloud

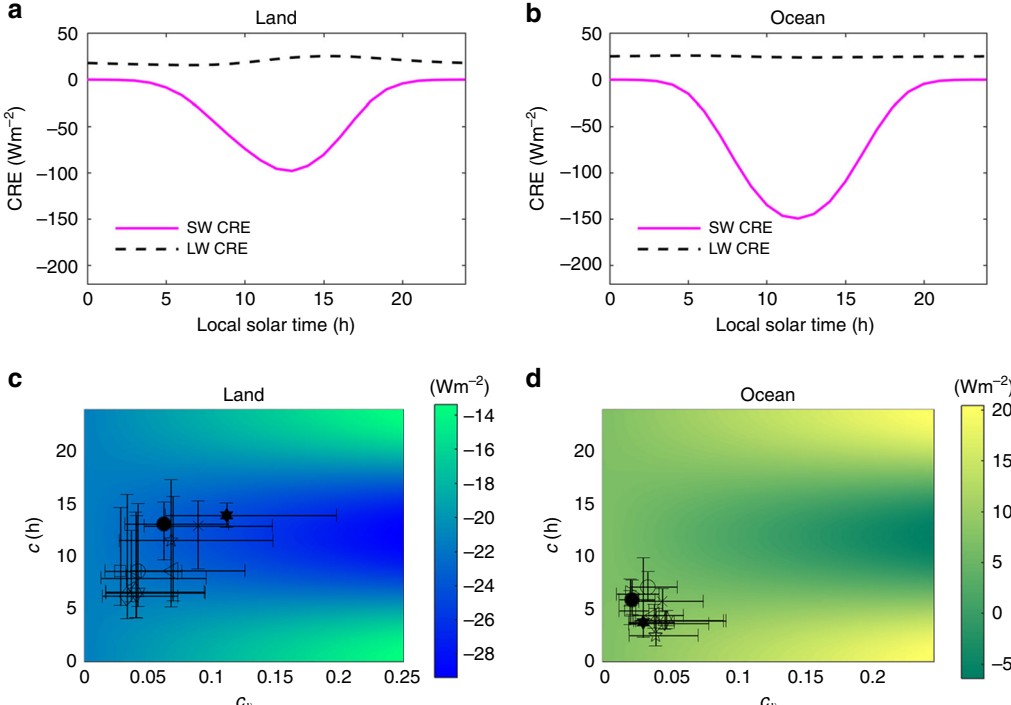

**Fig. 3** Cloud radiative effects and reference irradiance at the top of the atmosphere. Global diurnal cycle of cloud radiative effects (CRE) climatology over the land (**a**) and ocean (**b**) from reanalysis outputs separated into shortwave and longwave components; heatmap of TOA reference irradiance as a function of coefficient of variation and centroid of cloud diurnal cycles over the land (**c**) and ocean (**d**). The crosses specify the 25th, 50th, and 75th percentiles of the $c$ and $c_v$ from climate models (open markers, see details of the markers in Supplementary Table 1), satellite records (filled hexagram), and reanalysis data (filled circle)

radiative effects (CRE; see "Methods" section for details), which has been conventionally used to diagnose the effects of clouds by comparing all-sky and clear-sky radiative fluxes at the top of the atmosphere (TOA)[26–28].

We first analyze the diurnal cycle of global mean CRE at sub-daily timescale (see Eq. (10) in "Methods" section). We use data from ERA-20C reanalysis because its radiative flux data are similar to those from climate model outputs and satellite observations (see Supplementary Fig. 17). Figure 3a, b shows the diurnal cycle of CRE climatology over the land and ocean. The shortwave CRE, which is in phase with the incoming solar radiation, has a more marked diurnal variation than the longwave one. The CRE cycle is also stronger over the ocean due to the contrasting sea surface/cloud albedo that enhances the cloud effects. As explained in detail in the "Methods" section, we then use these CRE cycles to analyze the TOA reference irradiance as a function of the diurnal variations in cloud coverage. Such reference irradiance provides a consistent approach to evaluate the potential radiative impacts of the biases in DCC. Figure 3c, d shows a heatmap of the TOA reference irradiance as a function of the DCC indices $c$ and $c_v = \sigma/\mu$ for a sinusoidal form of diurnal cloud coverage (see Eq. (15) in "Methods" section). The reference irradiance is symmetric with respect to the centroid at noon ($c = 12$ h) and has higher gradients over the ocean. As one would expect, the enhanced CRE cycle over the ocean (compare Fig. 3a, b), due to its lower surface albedo, increases the DCC radiative impacts. Moreover, earlier cloud phases (i.e., corresponding to values of $c$ before sunrise) inevitably induce warming effects as clouds trap radiation at night regardless of cloud type and structure; similarly, midday cloud peaks typically induce cooling effects as clouds usually reflect more solar radiation at noon. Such impacts of the phase ($c$) become more significant for larger relative amplitude ($c_v$) and over the ocean. For example, for $c_v =$

0.1, the reference irradiance over the land increases by 6.4 W m⁻² in response to a shift of the centroid from noon to midnight (i.e., for $c$ going from 12 to 24), while for $c_v = 0.2$, the increase of irradiance becomes 12.7 W m⁻² for the same centroid shift. The corresponding increases over the ocean are even larger (11.0 and 21.7 W m⁻²). These large changes over both land and ocean are consistent with the values reported in a prior study[5] and are due to the significant and systematic variations of DCC.

**Radiative effects of cloud cycle errors**. To assess the radiative impacts of DCC errors, we first superimpose the $c$ and $c_v$ from different GCMs onto the heatmap in Fig. 3c, d. Over the land, the indexes appear much more scattered due to larger discrepancies of both $c$ and $c_v$ among the data sources, as already illustrated in Fig. 2. When the continental clouds tend to peak in the afternoon, as observed in ISCCP and simulated in ERA-20C and CNRM-CM5, they reflect more solar radiation and result in climates corresponding to the cold zone of the heatmap (Fig. 3c). Over the ocean, the indexes $c$ and $c_v$ are much more clustered. This however does not necessarily imply small radiative impacts, because the marine heatmap has steeper gradients (Fig. 3d). In summary, Fig. 3c, d shows potentially strong effects of DCC cloud errors in GCMs in both phase ($c$) and variability ($c_v$).

By focusing on the departure of cloud coverage from its mean, $f(t) = \mu + f_{DCC}(t)$, we can isolate the $f_{DCC}$ effects, without the sinusoidal approximation that was necessary for Fig. 3 (see Eq. (14) in "Methods" section). Accordingly, we calculate the TOA reference irradiance at each grid point in each season using $\mu$ from the ERA-20C reanalysis data and the $f_{DCC}$ from each GCMs outputs. In this way, $\mu$ from ERA-20C reanalysis is set as the baseline to compare the radiative impacts of $f_{DCC}$ from the climate models in terms of global mean TOA reference

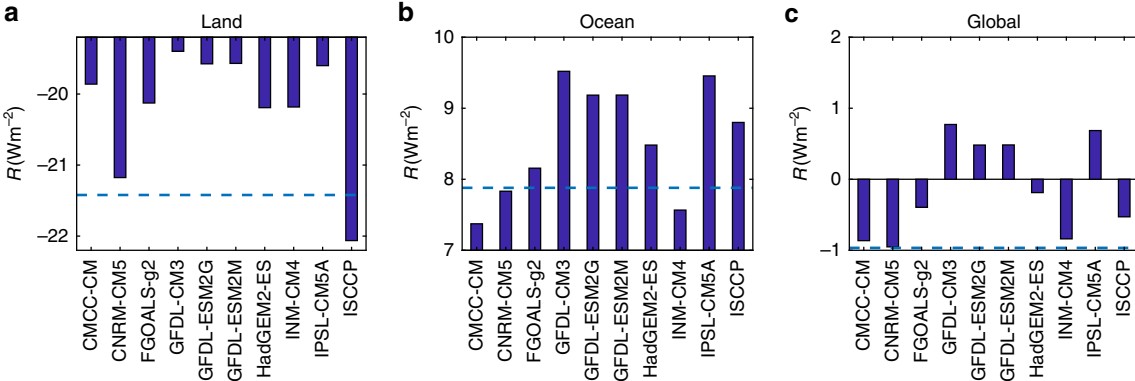

**Fig. 4** Reference irradiance. Top-of-the-atmosphere reference irradiance over **a** land, **b** ocean, and **c** whole Earth calculated using the mean cloud coverage ($\mu$) from reanalysis data and the departure of mean cloud coverage ($f_{DCC}$) from each data source. The dash lines show the top-of-the-atmosphere reference irradiance from reanalysis data. Note that the reanalysis is not designed to conserve the energy balance[48, 49]; therefore, the deviations from the standard reanalysis, rather than their absolute values, are more meaningful to quantify the radiative impacts of the DCC

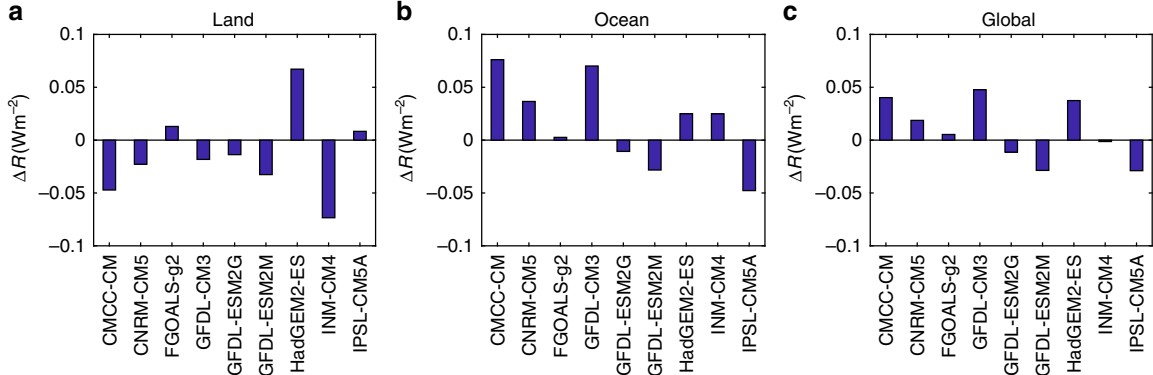

**Fig. 5** Change in reference irradiance in response to climate change. Change of TOA reference irradiance over **a** land, **b** ocean, and **c** whole Earth from 1986–2005 to 2018–2100. The 1986–2005 reference irradiance is calculated with GCM outputs from historical experiment as reported in Fig. 4, whereas the 2081–2100 irradiance is calculated in the same approach but with climate model outputs from RCP45 experiment

irradiance. The results, displayed in Fig. 4, show that over the land the lack of cloud peaks around afternoon in most GCMs (see Fig. 1) implies more solar radiation, so that the reference irradiance is higher than that from the standard ERA-20C. The inter-model difference of reference irradiance can be as large as 1.8 W m$^{-2}$ between climate models CNRM-CM5 and GFDL-CM3, and reaches the maximum 2.7 W m$^{-2}$ between GFDL-CM3 and ISCCP. Over the ocean, the relatively smaller DCC discrepancies (see Fig. 2) are amplified by their larger impacts (see the larger CRE over ocean in Fig. 3b), thus again resulting in considerably large inter-model differences in reference irradiance. The largest one occurs between CMCC-CM and GFDL-CM3 with a difference of 2.1 W m$^{-2}$. In CMCC-CM and INM-CM4, the surplus reference irradiance due to the lack of cloud peaks around noon over the land is somewhat compensated by the effects of the slightly later cloud peaks over the ocean. Other GCMs instead have a larger reference irradiance that is likely compensated by model tuning. Regarding the ISCCP records, it is lower over the land and higher over the ocean when compared with ERA-20C. An in-depth investigation of the DCC discrepancies between the multiple satellite observations and reanalyses is outside the scope of the work here, but it is worth mentioning that it could be useful as a starting point to formulate a standard DCC to be used as a reference for cloud parametrization in climate models. To verify that these results are robust to the selection of the baseline $\mu$, we also obtained it using data available from the CanAM4 climate model (this is the only GCM with sub-daily CRE outputs at the

global scale). As shown in Supplementary Fig. 18, the results are similar to the ones with ERA-20C, thus confirming our findings.

**Implication of cloud cycle errors for climate projection.** Since the total radiative effects of DCC biases may be very important, as shown by the previous analysis, it is likely that the GCM tuning, which is done to be able to reproduce the observed surface temperature climatology, may be in part linked to the DCC biases. It is thus crucial to try to understand the related consequences for climate predictions. First of all, climate models may have poor performance in simulating not only DCC, but also other cloud variables, such as structure and liquid water path[29, 30]. The errors in these climate variables may also interact and induce substantial biases on the radiation balance. For example, the well-documented problem of "too few, too bright," whereby the underestimation of cloudiness is often compensated by an overestimation of cloud albedo[31, 32], may further enhance the CRE[33–35], thus increasing the radiative effects of DCC errors. Furthermore, the overall radiation budget may be achieved by different tunings[36, 37] (e.g., on multiple climate variables or parameters in different times or locations). For instance, the "too few, too bright" problem may be also be compensated by adjusting the cloud structure and its microphysics[31, 34]. Similarly, as discussed above, the effects of large DCC errors over the land may be compensated by the opposite effects of small DCC errors over the ocean in some climate models (e.g., see Fig. 4).

To specifically assess the potential impacts of DCC errors on climate projection, we first consider the DCC changes from current to future simulations. Following the same approach as in Fig. 4, we calculate the future TOA reference irradiance with GCM outputs from RCP45 experiment during 2081–2100. The differences between future and present TOA reference irradiance ($\Delta R$), averaged over the land, ocean, and the whole Earth, are summarized in Fig. 5. As can be seen, $|\Delta R| \ll |R|$, indicating that the DCC responses to climate change have much smaller radiative impacts than its errors do in the current climate models. As a result, each GCMs model maintains consistent (albeit affected by errors) DCC simulations in future climate conditions. Thus, on the one hand, consistent biases in DCC between present and future climates give rise to similar TOA reference irradiance, so that the model tuning made for current climate conditions still remains largely effective for the global mean temperature projections. On the other hand, consistent biases have the potential to increase the uncertainty of climate projections. In fact, model tuning for extra TOA radiation is primarily conducted by adjusting cloud-related parameters[37], which may result in overestimation of CRE[33–35]. A large CRE likely strengthens the absolute values of cloud feedbacks[38, 39] and thus contributes to the large spread of climate projection among different GCMs. Moreover, while the effects of large DCC errors over land are compensated by the effects of small bias over the ocean, this compensation disrupts the spatial patterns of the energy distribution and may influence the land–ocean–atmosphere interaction, with potentially significant impacts on the climate projections[40]. It is therefore likely that improving the resolution of most climate model simulations, so that atmospheric convections are better resolved or at least more easily parameterized, will significantly improve DCC simulations[8, 10, 41]. This might also be the reason for the good DCC results of CNRM-CM5.

## Discussion

We have investigated the radiative effects of DCC errors in terms of total cloud coverage without identifying specifically the impacts of the diurnal cycle errors of in-cloud properties (e.g., cloud vertical structure, optical depth, and liquid/ice water path), which are all critical to the Earth's energy budget[1]. For example, while low clouds usually have higher cloud-top temperature and thus emit more longwave radiation, higher clouds emit less longwave radiation due to their lower temperature[42]. As shown in Fig. 3 and another prior independent study[13], the longwave CRE has much weaker cycle than its shortwave counterpart, indicating that DCC modulates Earth's energy primarily through the shortwave radiation. This suggests that the diurnal cycle errors of cloud structure will tend to have limited longwave radiative impacts (note that this should not be confused with the daily mean errors of cloud structure, which have long been recognized[1, 42] to have significant impacts on the Earth's energy balance).

Differently from the longwave radiation, the marked cycle of shortwave CRE (Fig. 3a, b) may indeed be influenced by cloud properties beyond the cloud coverage on which we focused here. For example, in-cloud water path, liquid/ice water content, and aerosol can influence the cloud albedo and thus adjust the CRE cycles[43, 44]. Once the GCMs provide detailed in-cloud properties in each grid point at sub-daily timescale, these potential impacts could be easily investigated with a similar approach to the one adopted here.

In summary, we have quantified the discrepancies of the DCC among current climate models, satellite observations, and reanalysis data. In general, climate models have better and more consistent performance in simulating mean cloud coverage, while most GCMs present considerable discrepancies in the standard deviation ($\sigma$) and centroid ($c$) of cloud cycles. The evident errors are the smaller $\sigma$ and earlier $c$ over the land, leading to an overestimation of net radiation as indicated by the CRE analysis. The smaller errors over the ocean also induce significant radiative impacts, as its relatively larger marine CRE amplifies the effects of DCC errors. Model tuning used to compensate for these errors results in shifts of the DCC phase over the ocean and even larger DCC biases over the land. Thanks to the limited responses of DCC to global warming, such biases do not seem to invalidate future climate projection; however, they may induce an overestimation of cloud-feedback strength and distort the patters of land–ocean–atmosphere interaction. Improving resolution and parameterizations of atmospheric convection may help reduce the reliance of model tuning and provide more accurate climate projections.

## Methods

**DCC.** The time series of cloud coverage at each grid box ($i$) in each season ($j$) from each data source ($m$) were analyzed as follows. For the period 1986–2005, in each day the cloud coverage is given at 3-h interval (e.g., at local solar time $t_1 = 3$ h, $t_2 = 6$ h,…,$t_k = 3 \cdot k$ hr,…,$t_8 = 24$ h). We first average, by season, these series to obtain a typical DCC coverage,

$$\bar{f}_{mij}(t_1), \quad \bar{f}_{mij}(t_2), \quad \dots \quad \bar{f}_{mij}(t_k), \quad \dots \quad \bar{f}_{mij}(t_8), \tag{1}$$

where subscripts $m$, $i$, $j$, and $k$ represent the data source index, grid location index, season index, and discrete time index, respectively. To characterize climatology of DCC, we define three indexes: the mean, amplitude, and phase.

The mean of the DCC is directly defined as the expectation

$$\mu_{mij} = \frac{1}{8} \sum_{k=1}^{8} \bar{f}_{mij}(t_k). \tag{2}$$

The amplitude of the DCC is quantified by its corrected sample standard deviation as:

$$\sigma_{mij} = \sqrt{\frac{1}{7} \sum_{k=1}^{8} \left[ \bar{f}_{mij}(t_k) - \mu_{mij} \right]^2}. \tag{3}$$

The coefficient of variation can be expressed as

$$(c_v)_{mij} = \frac{\sigma_{mij}}{\mu_{mij}}. \tag{4}$$

The latter is useful to analyze the impact of relative amplitude of DCC across different models.

The phase of the DCC is given by the centroid of $t_k$ weighted by the probability distribution of cloud coverage during one diurnal cycle

$$p_{mij}(t_k) = \frac{\bar{f}_{mij}(t_k)}{\sum_{k=1}^{8} \bar{f}_{mij}(t_k)}. \tag{5}$$

Since $t_k$ within one diurnal period can be treated as a circular quantity, the calculation of centroid ($c$) uses the circular statistics[45],

$$c_{mij} = \frac{\tau}{2\pi} \arg \left[ \sum_{k=1}^{8} p_{mij}(t_k) \exp \left( \mathbf{i} \frac{2\pi t_k}{\tau} \right) \right]. \tag{6}$$

where $\mathbf{i}$ is the imaginary unit and arg[·] is the argument of a complex number. As can be seen in the examples of Fig. 1 and Supplementary Fig. 2, the centroid is located around the timing of the most cloudiness in one typical day.

The RMSD of $\mu$ between data source $m_1$ and $m_2$ is defined as:

$$R_\mu(m_1, m_2) = \sqrt{\frac{1}{JI} \sum_j \sum_i \left( \mu_{m_1 ij} - \mu_{m_2 ij} \right)^2}, \tag{7}$$

where $I$ and $J$ are the numbers of grid boxes and seasons considered in the calculation of the corresponding RMSD. Similarly, the RMSD of $\sigma$ is

$$R_\sigma(m_1, m_2) = \sqrt{\frac{1}{JI} \sum_j \sum_i \left( \sigma_{m_1 ij} - \sigma_{m_2 ij} \right)^2}, \tag{8}$$

and the RMSD of $c$ is

$$R_c(m_1, m_2) = \sqrt{\frac{1}{JI} \sum_j \sum_i \left( c_{m_1 ij} - c_{m_2 ij} + n\tau \right)^2}, \tag{9}$$

where $n$ is an integer and $\tau$ is the length of one diurnal cycle (24 h). The integer $n$ is properly chosen such that the centroid difference $(c_{m_1 ij} - c_{m_2 ij} + n\tau)$ is within $[-\tau/2, \tau/2]$.

**TOA reference irradiance**. The CRE are conventionally defined as the difference of TOA all-sky ($R$) and clear-sky ($R_{clr}$) net radiative fluxes[26–28],

$$\text{CRE} = R - R_{clr}. \tag{10}$$

This quantity can also be expressed in terms of cloud coverage[42],

$$\text{CRE} = f(R_{cld} - R_{clr}), \tag{11}$$

where $R_{cld}$ is the cloudy-sky radiative flux. Combining Eqs. (10) and (11), $R_{cld}$ can thus be expressed as:

$$R_{cld} = \frac{1}{f}[R - (1-f)R_{clr}]. \tag{12}$$

Since all-sky, clear-sky radiative fluxes, and total cloud coverage are often provided in GCM outputs, $R_{cld}$ can be calculated directly from GCM outputs from Eq. (12).

With known values of $R_{cld}$, it is now possible to recalculate the TOA radiative flux as a function of cloud coverage and its properties, by solving Eq. (12) for $R$,

$$R = fR_{cld} + (1-f)R_{clr}, \tag{13}$$

where all the variables $R$, $R_{cld}$, $R_{clr}$, and $f$ are time dependent. Specifically, with Eq. (13) we can use the $R_{clr}$ and $R_{cld}$ provided by GCMs or other data sources to analyze the impacts of diurnal variations of cloud coverage on TOA radiation. To conduct this analysis, we decompose the cloud coverage first into a mean $\mu$ and fluctuations $f_{DCC}$ around it

$$f(t) = \mu + f_{DCC}(t). \tag{14}$$

We may also approximate the latter with its first harmonic[14, 15]

$$f_{DCC} \approx \sqrt{2}\sigma\cos[w(t - c)], \tag{15}$$

to directly link the reference irradiance to the phase ($c$) and the amplitude ($\sigma$).

Next, we substitute the sinusoidal approximation Eqs. (14) and (15) into Eq. (13). With $\mu$, $R_{cld}$, and $R_{clr}$ from ERA-20C reanalysis and $\sigma$ and $c$ from the GCM outputs, we are thus able to isolate the relative impacts of DCC phase and amplitude from each climate model (e.g., Fig. 3). Alternatively, focusing on the overall effect of DCC fluctuations ($f_{DCC}$), without the sinusoidal approximations Eq. (15), we can substitute Eq. (14) directly into Eq. (13), with $\mu$, $R_{cld}$, and $R_{clr}$ obtained from ERA-20C reanalysis and $f_{DCC}$ from GCMs (see Fig. 4). Both these two versions of daily mean TOA irradiance computed from Eq. (13) are referred to as TOA reference irradiance in the main text.

It is worth mentioning here that $R_{cld}$ and $R_{clr}$ are obtained from ERA-20C for assessing TOA reference irradiance in Figs 4 and 5 of the main text (alternative results using CanAM4 outputs are shown in Supplementary Figs. 18 and 19). In this way, we follow an approach which is similar in spirit to the standard radiative kernel approach used for estimating climate feedbacks[46]. A single set of radiative kernels usually is good enough for assessing climate feedback from different GCMs[47]. Similarly, the selection of standard $R_{cld}$ and $R_{clr}$ also has limited impacts on their inter-model patterns of TOA reference irradiance, which have been used for assessing the DCC radiative impacts (e.g., compare the results from ERA-20C in Figs 4 and 5 and from CanAM4 outputs in Supplementary Figs. 18 and 19).

**Code availability**. Models used in this paper are available from the corresponding author on request.

**Data availability**. The ISCCP satellite records are available from NASA Atmospheric Science Data Center (http://isccp.giss.nasa.gov/). The ERA-20C reanalysis data can be obtained from the European Centre for Medium-Range Weather Forecasts (http://www.ecmwf.int). The climate model data can be downloaded from the fifth phase of the Coupled Model Intercomparison Project website (http://cmip-pcmdi.llnl.gov).

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

## Acknowledgements

We acknowledge support from the USDA Agricultural Research Service cooperative agreement 58-6408-3-027; and the National Science Foundation (NSF) grants EAR-1331846, EAR-1316258, FESD EAR-1338694, and the Duke WISeNet Grant DGE-1068871.

## Author contributions

A.P. and J.Y. conceived and designed the study. J.Y. wrote an initial draft of the paper, to which both authors contributed edits throughout.

## Additional information

**Competing interests:** The authors declare no competing financial interests.

