## [Peer Review File · Nature Communications]

Reviewer #1 (Remarks to the Author):

This is an interesting and potentially valuable study that sheds some new light on an important aspect of clouds in climate. The importance of the diurnal cycle of clouds on the radiation budget has been known for more than 3 decades (e.g., Minnis and Harrison, JCAM, 1984), but it apparently has not been a topic of much interest for the climate modeling community. The diurnal cloud cycle (DCC) for each region is defined in terms of the mean, centroid and standard deviation. Those quantities taken from each dataset are used to compare the amplitudes and phase of the DCC and as input for the simplified radiation model. This study uses ISCCP along with the ERA-20 reanalysis as references for the period 1986-2005. A simple radiative balance model is used, in an idealized situation, to estimate the impact of the diurnal cycle discrepancies on the temperature differences between the models and observations. The implicit assumption is that the surface temperature will reach an equilibrium value that depends on the diurnal variation of cloud fraction, all other things being equal. Significant differences are found in amplitude and phase between the observations/reanalysis and the model results. Those differences cause the equilibrium reference temperatures to be systematically higher than the temperature computed with the ISCCP data alone. The models make up for this temperature difference and produce temperatures similar to actual surface observations by increasing cloud water path. Thus, climate models should be improved to better reflect the reality of the DCC and ultimately reduce the mean LWP used in the models. I believe this study would encourage GCM developers to take a harder look at the DCC and try to improve their representations in the models.

I like the study and think it should be published, but after addressing some concerns. As the authors note, performing a comprehensive analysis of the impact of the diurnal cycle would be a difficult and extended task, so they have taken a simplified approach that, at a minimum, shows that there is some impact on the surface temperature due to not capturing the diurnal cycle. The problem with the study is that it may be too simplified and would certainly exaggerate the diurnal impact. This is something the authors do not discuss much. I would like to see the following items discussed and/or used to revise the analysis.

1) Cloud heights, water path, and fractions all change over the diurnal cycle altering the radiation field. It is not just cloud fraction.

2) Assumptions used in the calculations: 0.06 for surface albedo everywhere. That might be ok for ocean, but land albedos are generally much greater. What is the sensitivity to using larger land albedos? Also, surface albedos typically increase with solar zenith angle, doubling for ocean. Different heat capacities were used for land and ocean, why not albedo?

3) Only LWP is used in the radiative study. How about using something more realistic such as total water path, or for that matter, the ISCCP total water path for each region? Ice water path is significant in many areas, especially where convection is dominant, and therefore should be included.

4) The assignation of the liquid water path overestimation as the main culprit in mitigating the differences in reference temperatures is only one factor. Ice water path is also a big factor and source of observations and GCM differences (Waliser et al., JGR 2009). Cloud heights and emissivities may also be important. These other potential error sources should be discussed.

5) Recognition was given to the long-term trend uncertainties in the ISCCP dataset, but there was

no discussion of the quality of the diurnal cycles produced by ISCCP. Has there been some assessment of the ISCCP diurnal cycles? Which data were used: VIS-IR or IR only?

The title is somewhat misleading. It implies that GCMs in general have some hidden warming. That could be inferred as excess warming in the trends or that models are running hot as a result of errors in cloud cycles. That is not really what is presented here. A more accurate title would be something like "Potential surface temperature impact of cloud diurnal cycle errors in climate models".

Reviewer #3 (Remarks to the Author):

This paper tried to investigate GCMs' deficiencies in representing the diurnal cycle of clouds and their effects on climate model simulations. While this GCMs' cloud bias can be an important topic, the authors missed their opportunities in addressing the key issue, i.e., how the bias might affect the fidelity of GCMs' simulation of climate variability and projections of future changes. In addition, their analysis approaches did not account for other cloud properties including GCM simulated cloud radiative effects, which are the most consequential quantities of the clouds. This makes their results less useful and hard to connect to the simple model. The conclusion on GCMs' overestimation of cloud liquid water path is both speculative and not very useful. Based on the above, I suggest the paper be rejected for publications in Nature Communications. The paper could be considered for resubmission if the major concerns below are satisfactorily addressed.

Major concerns:

1) It is well known that GCMs have large deficiency in representing the diurnal cycle of precipitation and clouds. While studies of GCMs' bias on precipitation and clouds are important, they are themselves not a good topic to be published in Nature Communications. However, the implications of these biases to the fidelity of the models' simulations of climate variability and projections of future changes would be appropriate to be published in Nature Communications. However, the key finding of this paper as highlighted in the bold text is "We show that, to compensate for the increased net radiation input implied by such errors, an overestimation of the cloud liquid water path may be introduced during calibration of 17 climate models ..." This is just one example of GCMs' compensation errors resulting from model tuning. The key question one would be interested in is how this bias might affect the fidelity of climate projection such as the cloud feedback and climate sensitivity. However, the authors missed their opportunities to address the most important question.

2) While the authors recognize clearly that clouds are important because of their radiative effect, the paper did not analyze any output of the GCM simulated cloud radiative effects (CREs), which are readily available and key to understand the climate simulations. Since CREs are results of many different properties of clouds

(amount/coverage, liquid/ice water content, optical properties, and their vertical distributions), it is not sufficient to understand CRE by just analyzing one aspect of the cloud properties. However, this paper took an extremely simplistic view of clouds and ignored entirely these important distinctions. In fact, this paper paid little attention in describing what GCM cloud variables are used. It only mentioned that cloud coverage was analyzed. But there are a number of different measures of the cloud coverage in GCMs (e.g., low, middle, high, and total cloud cover). Low and high clouds might exhibit different diurnal cycle and affect radiative transfer in different ways. Many GCMs also produce cloud covers using satellite simulators. Despite all of the possible complications, I did not find even a full sentence to describe this. The missing of the analysis of GCM simulated CRE make the readers hard to imagine the actual diurnal cycle of CRE in the models. It also makes it hard to connect the GCMs with the simple radiative balance model. Why should one trust the extremely simple radiative balance model if it is not demonstrated to be relevant to the comprehensive GCMs' results?

3) Based on the authors' metrics, the CNRM-CM4 model outperform most other GCMs in simulating the diurnal cycle of clouds. This provides a very good opportunity for the authors to investigate the implication and consequence of the diurnal cycle of clouds on climate projections since the contrast between good and bad diurnal cycle models would help to identify key roles of the diurnal cycle of clouds in climate simulation and projections. I hope the authors can pursue this.

4) The GCM clouds are typically tuned to produce the observed global TOA radiative fluxes for both total and individual (LW and SW) components. This tuning is typically done in AMIP simulations forced by observed SSTs. Since the global surface temperature is largely constrained by the specified SSTs, it is not really involved in calibrating the GCM clouds. Moreover, there are many different ways GCM clouds can be tuned to produce the correct total flux. Liquid water path is only one of them. Thus, the authors' conclusion of overestimation of LWP is speculative. It is also not very useful because it is well known about the compensation errors in tuning clouds. The question is what is the real consequence of the compensation errors. Does it matter for climate projections? Why or Why not?

Minor comments:

There are a number of locations in the paper that miss proper definitions, explanations or references. Below are a few examples:

Line 37: What are exactly the cloud coverage variable used from the models? Do you distinguish between low, middle, and high cloud? Do you use products from satellite simulator?

Line 40-42: Please provide evidence or reference that the use of centroid defined in this paper compares well with other methods in determining the phase of diurnal cycle.

Line 51-52: Why CNRM-CM4 relies on ECMWF IFS?

Line 104-106: This is what I hope the authors to focus on. I am disappointed that the authors did not pursue this further.

Line 108-109 and line 313-348: The section of description of the radiative balance model lacks of explanation, references, and validation for the formulations, parameter setting, and model performance. It is not clear to what extent it is relevant for understanding the GCM simulated effect of diurnal cycle of clouds since no GCM results are compared.

Line 149-152: GCM clouds are typically tuned to produce correct global TOA fluxes in simulations forced by observed SST. Moreover, there are many ways to tune the clouds, LWP is only one of them.

Line 156-157: The correlation is not very good. Even it is good, it may be due to different reasons (e.g., cloud phase response to surface temperature).

Line 159-161: Why not directly examine the diurnal cycle of TOA radiative flux or cloud radiative effects?

Line 174-184: The paper does not have a strong conclusion.

Reviewer #4 (Remarks to the Author):

This manuscript documents biases in the diurnal cycle of clouds in the CMIP5 model ensemble. The authors first claim that while the mean cloud cover in most models is fairly well captured, the amplitude and the phase of the diurnal cycle exhibit important biases. Over land, the amplitude is too small and the cloud cover peaks in the morning rather than in the afternoon. The oceanic diurnal cycle is better captured, but cloud cover still peaks too early in the night. The second part of the manuscript tries to explain the importance of these biases from a radiation point of view, by applying a minimalist radiative transfer model. This analysis reveals that the biases in the cloud diurnal cycle should lead to an excess radiation at the surface and in turn a very significant warm bias in all models. However, models tend to compensate for the lack of cloud by having too large in-cloud water contents, so that the average global radiative impact of clouds in most models is still fairly well captured.

The study of the diurnal cycle of clouds is important and to my knowledge this is the first study tackling this issue on a global scale. The method of decoupling the mean, amplitude and phase of the diurnal cycle seems sound, but I do wonder why the typical noon-peak in cloudiness over land in climate models is absent in their extended data figure 2 in most models. I wonder if the authors could comment on this. Furthermore, many studies on the diurnal cycle of clouds and precipitation in specific regions have been done before and as such, this study is not presenting an entirely surprising or novel result. Studies over the past 15 years on the diurnal cycle of clouds and precipitation in climate models include: Yang and Slingo (2001), Clark et al. (2007), Pfeifroth et al. (2012), Langhans et al. (2013), Walther et al. (2013) or Gustafson et al. (2015). These studies recognise for various regions on the globe that the cloud diurnal cycle is poorly captured in climate models, but that this greatly improves when these models are run at convection-permitting scales (e.g. Clark et al. 2007, Langhans et al. 2013, Brisson et al 2016), suggesting the parameterization of convection is the main culprit for this deficiency.

Moreover, the claim by the authors that the lack of cloudiness in the daytime is compensated by

too large in-cloud liquid water content ties in with the well-documented 'too few, too bright' problem that has been extensively studied as well, even in the context of the CMIP5 models (e.g. Williams and Webb 2009, Nam et al. 2012, Tsushima et al. 2015).

While I think there is certainly merit in the first part of their analysis, I'm not convinced by the minimalistic radiative transfer model approach to show the possible impact of the cloud biases on the radiation balance. The authors come up with a simple 1D radiative transfer model and plug in the globally averaged values for cloud albedo and water path in their calculations. Furthermore, they apply this model on a diurnal cycle of cloud cover including their observed and simulated values of the mean, amplitude and phase. In their supplementary figures, they show large variations in these three diurnal cycle statistics, not necessarily with a good correlation between the biases in these three parameters. Given that radiative transfer is a highly non-linear process, I feel that this 1D radiative transfer model bears very little resemblance to what the real impact of cloud biases in the CMIP5 models on their global radiation balance would be. Indeed, it matters a lot where biases in amplitude and phase are overlapped or where the biases in water path and amplitude are overlapped. Furthermore, an important simplification is the fact that the authors do not discriminate between high clouds and low clouds in their analysis. High clouds are known to have a predominant warming effect, while liquid clouds have a predominant cooling effect on earth's climate (Ramanathan et al. 1989). By simply assuming a globally averaged cloud albedo and water content, this important distinction between ice and liquid clouds is ignored. Hence, it is very likely that the suggested very large impact of the CMIP5 bias in the cloud diurnal cycle on the radiation balance is incorrect. This is also reflected in the fact that the simple radiative transfer model only reaches a mean global temperature of 285.5 K (Figure 4), which is 2-3 K colder than the observed global mean temperature (or even lower, given that the authors omit the poles from their analysis).

Last, I have a minor comment about Figure 2. I'm not sure whether these contingency tables are the best way to convey the message here. Wouldn't it be better to just show bar plots of the mean observed (ISCCP and ERA) and each of the CMIP modelled mean cloud cover, amplitude and phase? Or alternatively, just bar plots of the RMSD between the observations and each of the CMIP models. I'm not sure what we learn from the RMSD between each of the different models.

Based on the fact that the first part of this paper is relevant and well executed, but not entirely novel, and that the second part of the analysis seems to be a too simple approach to support their fairly far-reaching conclusion, I would not recommend this paper to be accepted in its current form. I think a far more advanced approach is required to show the impact of current cloud biases in climate models on the radiation balance.

References:

Brisson, E., Van Weverberg, K., Demuzere, M. et al. (2016). How well can a convection-permitting climate model reproduce decadal statistics of precipitation, temperature and cloud characteristics? *Clim Dyn* 47: 3043.

Clark, A. J., Gallus, W., Chen, T.-C. (2007). Comparison of the diurnal precipitation cycle in convection-resolving and non-convection-resolving mesoscale models. *Monthly Weather Review*. 135, 3456-3473.

Gustafson Jr., W.I., Ma, P.-L., Singh, B. (2015). Precipitation characteristics of CAM5 physics at mesoscale resolution during MC3E and the impact of convective timescale choice. *Journal of Advances in Modeling Earth Systems* 6, 1271-1287.

Langhans W, Schmidli J, Fuhrer O, Bieri S, Schär C (2013). Long-term simulations of thermally-

driven flows and orographic convection at convection-parameterizing and cloud-resolving resolutions. *J Appl Meteorol Climatol* 1:130117155938002 (1984)

Nam C, Bony S, Dufresne J, Chepfer H (2012). The 'too few, too bright' tropical low-cloud problem in CMIP5 models. *Geophys Res Lett*. doi:10.1029/2012GL053421

Pfeifroth U, Hollmann R, Ahrens B (2012). Cloud cover diurnal cycles in satellite data and regional climate model simulations. *Meteorol Zeitschrift* 21(6):551–560

Ramanathan, V., R. D. Cess, E. F. Harrison, P. Minnis, B. R. Barkstrom, E. Ahmad and D. Hartmann (1989). Cloud-Radiative Forcing and Climate: Results from the Earth Radiation Budget Experiment. *Science New Series*, 243, 57-63

Tsushima Y., Ringer, A.M., Tsuyoshi, K., Kawai, H., Roerhig, R., Cole, J., Watanabe, M., Yokohata, T., Bodas-Saledo, A., Williams, K., Webb, M. (2015). Robustness, uncertainties, and emergent constraints in the radiative responses of stratocumulus cloud regimes to future warming, *Clim Dyn*, DOI 10.1007/s00382-015-2750-7

Walther, A., Jeong, J.-H., Nikulin, G., Jones, C., Chen, D. (2013). Evaluation of the warm season diurnal cycle of precipitation over Sweden simulated by the Rossby Centre regional climate model RCA3. *Atmospheric Research*. 119, 131-139.

Williams K, Webb M (2009). A quantitative performance assessment of cloud regimes in climate models. *Clim Dyn* 33(1):141–157.

Yang, G.Y., Slingo, J. (2001). The diurnal cycle in the Tropics. *Monthly Weather Review* 129, 784-801.

Each reviewer comment (italicized) is followed by a response.

Reviewer #1:

This is an interesting and potentially valuable study that sheds some new light on an important aspect of clouds in climate.

We thank the reviewer for his/her positive and constructive comments and encouragement.

The importance of the diurnal cycle of clouds on the radiation budget has been known for more than 3 decades (e.g., Minnis and Harrison, JCAM, 1984), but it apparently has not been a topic of much interest for the climate modeling community. The diurnal cloud cycle (DCC) for each region is defined in terms of the mean, centroid and standard deviation. Those quantities taken from each dataset are used to compare the amplitudes and phase of the DCC and as input for the simplified radiation model. This study uses ISCCP along with the ERA-20 reanalysis as references for the period 1986-2005. A simple radiative balance model is used, in an idealized situation, to estimate the impact of the diurnal cycle discrepancies on the temperature differences between the models and observations. The implicit assumption is that the surface temperature will reach an equilibrium value that depends on the diurnal variation of cloud fraction, all other things being equal. Significant differences are found in amplitude and phase between the observations/reanalysis and the model results. Those differences cause the equilibrium reference temperatures to be systematically higher than the temperature computed with the ISCCP data alone. The models make up for this temperature difference and produce temperatures similar to actual surface observations by increasing cloud water path. Thus, climate models should be improved to better reflect the reality of the DCC and ultimately reduce the mean LWP used in the models. I believe this study would encourage GCM developers to take a harder look at the DCC and try to improve their representations in the models.

We are glad that the reviewer finds our approach novel and worthwhile. We thank the reviewer for highlighting nicely the most novel parts of our work. We have used some of these comments to improve the presentation of the revised version.

I like the study and think it should be published, but after addressing some concerns. As the authors note, performing a comprehensive analysis of the impact of the diurnal cycle would be a difficult and extended task, so they have taken a simplified approach that, at a minimum, shows that there is some impact on the surface temperature due to not capturing the diurnal cycle. The problem with the study is that it may be too simplified and would certainly exaggerate the diurnal impact. This is something the authors do not discuss much. I would like to see the following items discussed and/or used to revise the analysis.

We agree with the reviewer and acknowledge the potential shortcomings of using a minimalist model. This issue has also been raised by the other reviewers. To address and solve the problem at its roots, in the updated manuscript, we decided to re-do all the analyses, by extending the cloud radiative effects (CRE) method and using it to estimate the TOA radiative fluxes and analyze the radiative impacts of cloud cycle biases. We explain these extensions, used instead of the minimalist radiative model, in the revised Method section.

The analysis of the DCC impacts are now carried out consistently by calculating a reference radiation based on CRE. To do so, we use the definition of cloud radiative effects (CRE) (Cess et al. 1989, 1990, 1996),

$$\text{CRE} = R - R_{\text{clr}}, \quad (1)$$

where R and R_{clr} are all-sky and clear-sky radiative fluxes. This quantity can also be expressed in terms of cloud coverage as in Eq. 4 of Ramanathan et al. (1989),

$$\text{CRE} = f(R_{\text{cld}} - R_{\text{clr}}), \quad (2)$$

where R_{cld} is the cloudy-sky radiative flux. Combining (5) and (6), R_{cld} can thus be expressed as

$$R_{\text{cld}} = \frac{1}{f}[R - (1 - f)R_{\text{clr}}]. \quad (3)$$

Since all-sky, clear-sky radiative fluxes, and total cloud coverage are often provided in GCM outputs, R_{cld} can be calculated directly from GCM outputs using (3).

With known values of R_{cld} , it is now possible to recalculate the TOA radiative flux as a function of cloud coverage and its properties, by solving (3) for R ,

$$R = fR_{\text{cld}} + (1 - f)R_{\text{clr}}. \quad (4)$$

With (4), we can use the R_{clr} and R_{cld} provided by GCMs or other data sources to analyze the impacts of diurnal variations of cloud coverage on TOA radiation. The radiation R in this analysis is referred to as TOA reference radiation in the text and is used to analyze the radiative impacts of DCC errors.

We have extensively revised the manuscript to incorporate and explain these changes made to avoid potential biases or excessive simplifications introduced by the minimalist model.

1) Cloud heights, water path, and fractions all change over the diurnal cycle altering the radiation field. It is not just cloud fraction.

We agree. In the updated manuscript, we now use CRE and TOA radiation from the state-of-the-art ERA-20C reanalysis, which takes into account the diurnal cycle of all the important cloud properties (e.g. cloud heights, water path, liquid/ice water content).

On the other hand, the reviewer's question raises other interesting questions. Are there diurnal cycle biases of other cloud properties? Are the radiative impacts of these biases important? These questions are interesting but can only in part be addressed with the current GCM outputs. The issues are now discussed in the last section of the revised manuscript.

In particular, the heights of the clouds roughly determine the cloud-top temperature and primarily influence the longwave radiation. In one independent study from Webb et al. (2015), the authors have shown that the longwave CRE have much smaller diurnal cycles than their shortwave counterpart (Figures 2c and 4c in their paper as also reported here in Figure 1). This means that diurnal cycles of clouds adjust the energy balance primarily through their modulation on shortwave radiation. Therefore, even if there are diurnal cycle errors of cloud heights among climate models, these errors should have much smaller longwave radiative impacts. Note that a weak longwave CRE cycle only suggest that the **diurnal cycle errors of cloud height** are less important, while obviously the **daily mean cloud height errors** could still have significant radiative impacts.

We also acknowledge, in the revised manuscript, that diurnal cycle of cloud heights can change the cloud-top temperature and consequently induces the change of liquid/ice water content. In this way, diurnal cycle errors of cloud heights also influence the shortwave radiation and need to be analyzed.

With regards to the in-cloud water path, it may also have certain diurnal cycle errors that necessarily influence the cloud albedo. This will likely contribute to the SW CRE. While these effects are worth being analyzed, we believe they are outside the scope of the manuscript.

More importantly, the novel approach provided in this paper can now be adopted to analyze the impacts of diurnal cycle errors of liquid/ice content and in-cloud water path once the groups participating in the CMIP5 project provide the values of the essential in-cloud properties at sub-daily timescale over the globe. Currently, they only provide these values in ‘AMIP’ experiments at approximately 120 ‘cfSites’ (Taylor et al. 2011; Bony et al. 2011; Webb et al. 2015), which are not sufficient to assess global conditions. We noted this in the updated manuscript and pointed to the generality of our approach.

Figure 1. Diurnal cycle of cloud radiative effects (CRE) in AMIP experiment from climate models. After Webb et al. (2015).

2) Assumptions used in the calculations: 0.06 for surface albedo everywhere. That might be ok for ocean, but land albedos are generally much greater. What is the sensitivity to using larger land albedos? Also, surface albedos typically increase with solar zenith angle, doubling for ocean. Different heat capacities were used for land and ocean, why not albedo?

The reviewer is correct: the albedo over the ocean is smaller and is influenced by the solar zenith angle. In the updated manuscript, we directly used the radiative fluxes from ERA-20C reanalysis to study the radiative impacts of cloud cycle errors so this problem is now solved. The new results show that the diurnal cycles of CRE climatology over the ocean are stronger than those over the land (see Figure 2 below), which is consistent with the reviewer’s comments.

It is also interesting to note that the CRE from ERA-20C is similar in magnitude to those from ‘AMIP’ experiment at approximately 120 ‘cfSites’ as reported by Webb et al. (2015) (see Figure 1 above).

Figure 2. Global average diurnal cycle of cloud radiative effects (CRE) from ERA-20C reanalysis during 1986-2005 over land and ocean.

3) *Only LWP is used in the radiative study. How about using something more realistic such as total water path, or for that matter, the ISCCP total water path for each region? Ice water path is significant in many areas, especially where convection is dominant, and therefore should be included.*

We followed the reviewer's suggestion and used ERA-20C reanalysis data which account for more realistic cloud properties. We also assessed the CRE and TOA radiative fluxes at each grid in each season so that the seasonal and spatial variations of cloud properties are also considered.

Note that the consideration of more realistic cloud properties is not conflict with the answer to the reviewer's question 1. While we do consider all important cloud properties for the CRE and TOA radiative fluxes, we only analyze the radiative impacts of diurnal cycle biases of total cloud coverage. The potential impacts of diurnal cycle biases of in-cloud properties (e.g. liquid/ice cloud content, cloud structure) are now discussed in the last section of the revised manuscript (see previous answer).

4) *The assignation of the liquid water path overestimation as the main culprit in mitigating the differences in reference temperatures is only one factor. Ice water path is also a big factor and source of observations and GCM differences (Waliser et al., JGR 2009). Cloud heights and emissivities may also be important. These other potential error sources should be discussed.*

We agree with the reviewer that the analysis of liquid water path overestimation is somewhat speculative. Indeed, multiple climate variables may have errors and need to be tuned; multiple parameters can be adjusted to balance the Earth's radiation. This makes it hard to disentangle the precise error-adjustment mapping.

In the updated manuscript, we commented on the potential factors that may compensate the impacts of cloud cycle errors, as suggested by the reviewer. We also commented that the large DCC errors over the land can be compensated by the small DCC bias over the ocean in some climate models.

As a final and more interesting conclusion of this paper, we go a step further and analyze the implications of these DCC errors to the future climate projection. We found that DCC errors do not seem to invalidate future climate projection, although may likely induce overestimation of cloud-feedback strength and distort the patters of land-ocean-atmosphere interaction.

5) Recognition was given to the long-term trend uncertainties in the ISCCP dataset, but there was no discussion of the quality of the diurnal cycles produced by ISCCP. Has there been some assessment of the ISCCP diurnal cycles? Which data were used: VIS-IR or IR only?

A literature review shows that DCC climatology from ISCCP records is consistent with the observations from stationary weatherships and from other satellite records (Cairns 1995; Rozendaal et al. 1995; Wylie and Woolf 2002). In the updated manuscript, we followed Rozendaal et al. (1995) and used the IR channel which has been consistently measured throughout the whole diurnal cycle. The differences of diurnal cycle of clouds calculated from IR and VIS-IR are primarily located over the ocean. We have explained the data source and updated all the corresponding figures and text.

*The title is somewhat misleading. It implies that GCMs in general have some hidden warming. That could be inferred as excess warming in the trends or that models are running hot as a result of errors in cloud cycles. That is not really what is presented here. A more accurate title would be something like *ÉC;Potential surface temperature impact of cloud diurnal cycle errors in climate modelsÉD;**

Thanks for the suggestion. Since we have shifted our focus from temperature to the TOA radiative flux, we changed the title to “effects of cloud cycle bias in climate models”.

Reviewer #3 (Remarks to the Author):

This paper tried to investigate GCMs' deficiencies in representing the diurnal cycle of clouds and their effects on climate model simulations. While this GCMs' cloud bias can be an important topic, the authors missed their opportunities in addressing the key issue, i.e., how the bias might affect the fidelity of GCMs' simulation of climate variability and projections of future changes.

We thank the reviewer for his/her insightful and constructive comments. In general, we agree with the reviewer that the purpose of GCM is to provide accurate forecast of future climate conditions.

To more directly address the issues related to cloud bias on climate variability and projections, we performed further analyses and thoroughly revised our manuscript. In particular, we analyzed the cloud cycle in response to the climate change. The new analyses, detailed below and in the revised text, show that DCC may not significantly change under global warming. Especially the CNRM-CM5, with relatively accurate simulation of DCC, shows that its feedback strength is limited. Interestingly, this indicates that the model tuning conducted in the current climate conditions may still remain largely effective for the future climate projections.

We also discussed some indirect influences of DCC errors on the land-ocean interactions and cloud feedback strength, as detailed in our specific responses below.

In addition, their analysis approaches did not account for other cloud properties including GCM simulated cloud radiative effects, which are the most consequential quantities of the clouds. This makes their results less useful and hard to connect to the simple model.

We followed the reviewer's suggestion and extended the cloud radiative effects (CRE) to analyze the DCC radiative impacts (now explained in detail in the Methods section). We recalculated the TOA radiation as a function of cloud coverage, referred to as TOA reference radiation in the text, and used it to assess the radiative impacts of DCC errors. We have revised the manuscript to reflect these changes and explain these new results.

The conclusion on GCMs' overestimation of cloud liquid water path is both speculative and not very useful.

We agree. We have dropped this analysis of the overestimation of liquid water path. To pursue a more useful conclusion, we extended our analysis to consider both direct and indirect implications of these DCC errors to the future climate projection. We found that model tuning from current climate conditions may still remain largely effective for its future climate prediction due to the weak DCC feedback. However, it may induce an overestimation of cloud-feedback strength and distort the patterns of land-ocean-atmosphere interaction.

Based on the above, I suggest the paper be rejected for publications in Nature Communications. The paper could be considered for resubmission if the major concerns below are satisfactorily addressed.

We have made a very substantial revision of the manuscript. As explained above, we have used CRE and TOA radiative fluxes directly from reanalysis data to assess the radiative impacts of DCC errors. We have analyzed the cloud cycle in response to the climate change to explain how DCC may influence the future climate projection. We also discussed the indirect impacts of DCC errors to the cloud feedbacks and the spread in model climate projection. We hope this major revision satisfactorily addresses the reviewer's concerns and makes our manuscript worth of publication in Nature Communications.

Major concerns:

1) It is well known that GCMs have large deficiency in representing the diurnal cycle of precipitation and clouds. While studies of GCMs' bias on precipitation and clouds are important, they are themselves not a good topic to be published in Nature Communications. However, the implications of these biases to the fidelity of the models' simulations of climate variability and projections of future changes would be appropriate to be published in Nature Communications.

Motivated by the reviewer's suggestions, we analyzed the changes of cloud cycle from 'historical' experiment during 1986-2005 to 'RCP45' experiment during 2081-2100. We found that these changes are quite small in climate models and they have much smaller radiative impacts than those from cloud cycle biases (see the results in Figure 3 below). Particularly, CNRM-CM5 with its relatively accurate DCC simulation also suggests a somewhat limited DCC feedback. Therefore, the cloud cycles are quite consistent in current and future climate conditions, indicating that the model tuning conducted for current climatology may remain effective for climate projection in spite of DCC biases.

We also discussed potential indirect effects of DCC errors in the revised manuscript. As shown in Figure 3 (top panel), DCC errors from CMCC-CM and INM-CM4 over land induce extra TOA radiations, which are almost completely compensated by the slightly later cloud peak over the ocean. Consequently, the global mean TOA reference radiations are close to the standard ERA-20C state (the dash line). Other climate models may use specific parameters to adjust the surplus TOA radiations. Model tuning, in all the different ways it can be carried out, may have some impacts on the simulation of land-ocean-atmosphere interactions. Moreover, if the extra TOA radiations are adjusted by increasing cloud albedo similar to the well-documented 'too few, too bright' problem in climate models (Williams and Webb 2009; Nam et al. 2012; Tsushima et al. 2016), DCC errors may enhance the CRE and increase the absolute value of cloud feedbacks. The consequence of large cloud feedbacks is the increase of the spread in model climate projection (Karlsson et al. 2008; Brient and Bony 2012).

In summary, while DCC errors do not seem to invalidate future climate projection, they may cause an overestimation of cloud-feedback strength and distort the patterns of land-ocean-atmosphere interaction.

Figure 3. (Top panel) TOA reference radiation averaged in each grid point and each season over the land, ocean, and the whole Earth with mean cloud coverage from ERA-20C reanalysis and the diurnal fluctuation from each data source during 1986-2005. The dash lines show the TOA reference radiation from ERA-20C reanalysis. Note ERA-20C may induce some artificial energy sources (Berrisford et al. 2009; Dee et al. 2011) so that global mean radiation is around -0.9 Wm^{-2} . **(Bottom panel)** DCC responses to climate change in terms of change of TOA reference radiation. The current TOA reference radiations are from top panel, whereas the future radiations are calculated in the same approach but from ‘RCP45’ experiment during 2081-2100. The differences between current and future are the DCC radiative response to the climate change.

However, the key finding of this paper as highlighted in the bold text is "We show that, to compensate for the increased net radiation input implied by such errors, an overestimation of the cloud liquid water path may be introduced during calibration of 17 climate models ..." This is just one example of GCMs' compensation errors resulting from model tuning. The key question one would be interested in is how this bias might affect the fidelity of climate projection such as the cloud feedback and climate sensitivity. However, the authors missed their opportunities to address the most important question.

We agree with the reviewer that it is more important to analyze the role of cloud cycle in forecasting future climates. In the updated manuscript, we shifted our focus from the error compensation to the cloud cycle response to the climate change. As discussed above and reported in Figure 3, GCMs still have the ability to provide reasonable prediction after the model tuning but we need to be cautious that the DCC errors may increase the cloud feedbacks and distort the patterns of

land-ocean-atmosphere interactions. We believe this is an interesting finding. We are grateful for the suggestion.

2) While the authors recognize clearly that clouds are important because of their radiative effect, the paper did not analyze any output of the GCM simulated cloud radiative effects (CREs), which are readily available and key to understand the climate simulations. Since CREs are results of many different properties of clouds (amount/coverage, liquid/ice water content, optical properties, and their vertical distributions), it is not sufficient to understand CRE by just analyzing one aspect of the cloud properties. However, this paper took an extremely simplistic view of clouds and ignored entirely these important distinctions.

We thank the reviewer for pointing this out. We acknowledged that the minimalist model is not comprehensive enough to simulate the impacts of all types of cloud properties on earth energy budget. In the revised manuscript, we now directly use the CRE to analyze the radiative impacts of cloud cycle biases. As explained in Methods section of the revised manuscript, CRE is conventionally defined as the difference of TOA all-sky (R) and clear-sky (R_{clr}) net radiative flux (Cess et al. 1989, 1990, 1996),

$$\text{CRE} = R - R_{\text{clr}}. \quad (5)$$

This quantity can also be expressed in terms of cloud coverage as Eq. 4 in Ramanathan et al. (1989),

$$\text{CRE} = f(R_{\text{cld}} - R_{\text{clr}}), \quad (6)$$

where R_{cld} is the cloudy-sky radiative flux. Combining (5) and (6), R_{cld} can thus be expressed as

$$R_{\text{cld}} = \frac{1}{f}[R - (1 - f)R_{\text{clr}}]. \quad (7)$$

Since all-sky, clear-sky radiative fluxes, and total cloud coverage are often provided in GCM outputs, R_{cld} can be calculated directly from GCM outputs using (7).

With known values of R_{cld} , it is now possible to recalculate the TOA radiative flux as a function of cloud coverage and its properties, by solving (7) for R ,

$$R = fR_{\text{cld}} + (1 - f)R_{\text{clr}}, \quad (8)$$

Specifically, with (8) we can use the R_{clr} and R_{cld} provided by GCMs or other data sources to analyze the impacts of diurnal variations of cloud coverage on TOA radiation. This radiation used for analyzing DCC errors is referred to as TOA reference radiation.

Figure 4 shows both the diurnal cycle of CRE climatology and the corresponding TOA reference radiation as functions of centroid and standard deviation of cloud cycle.

Note, also that Figure 4 is derived from global averaged CRE to demonstrate how DCC influences TOA radiation. Another complementary analysis was conducted over each grid point in each season (e.g. see Figure 3). We have updated the methods and the corresponding results in the text to incorporate these new analyses.

Figure 4. CRE and TOA reference radiation. Diurnal cycle of CRE climatology over land (a) and ocean (b) from ERA-20C reanalysis; ‘heatmap’ of TOA reference radiation over the land (c) and ocean (d) as functions of coefficient of variation and centroid of cloud diurnal cycles. The crosses specify the 25th, 50th, and 75th percentiles of the c and c_v from GCMs (open markers), ISCCP records (filled hexagrom), and ERA-20C reanalysis data (filled circle). Note the 10th, 50th, and 90th percentiles of c and c_v over the ocean were shown in the original manuscript but has been changed to the consistent 25th, 50th, and 75th in the revised manuscript.

In fact, this paper paid little attention in describing what GCM cloud variables are used. It only mentioned that cloud coverage was analyzed. But there are a number of different measures of the cloud coverage in GCMs (e.g., low, middle, high, and total cloud cover). Low and high clouds might exhibit different diurnal cycle and affect radiative transfer in different ways. Many GCMs also produce cloud covers using satellite simulators. Despite all of the possible complications, I did not find even a full sentence to describe this.

In the updated manuscript, we directly use CRE and TOA radiative fluxes from the state-of-the-art ERA-20C reanalysis, which include the effect of all the important cloud variables (e.g. low, middle, and high clouds) and their respective diurnal cycles.

We now also discuss other interesting questions related to this point. For example, are there any cloud cycle biases of cloud structure? Do these biases also have strong impacts on the radiation

budget? We have discussed these questions in the last section of the revised manuscript. Cloud structure may affect radiative transfer in different ways. Typically, the low clouds have higher temperature, emit more longwave radiation, and tend to cool the surface; the high clouds may warm the Earth by emitting less longwave radiation. The cloud structure controls the radiation budget primarily through the longwave components. If the diurnal cycles of cloud structure are important, the corresponding longwave CRE will show strong diurnal variations. However, one independent study from Webb et al. (2015) has shown that the longwave CRE has much smaller diurnal cycle than its shortwave counterpart (see Figure 5 below). Our baseline CRE from ERA-20C also shows similar results (see Figure 4 above). These results explain that diurnal cycles of clouds adjust the energy balance primarily through its modulation on shortwave radiation. Therefore, even if there are certain diurnal cycle errors of cloud structure among climate models, their longwave radiative impacts are less important. Note that we only suggest that **diurnal cycle errors of cloud height** are less important, but obviously not that the **daily mean cloud height errors** do not have significant radiative impacts.

Cloud height roughly determines the cloud-top temperature, which may indirectly influence the liquid/ice water content and cloud albedo. In this sense, the indirect shortwave radiative impacts of diurnal cycles of cloud vertical structure may need further investigation. However, climate modelling groups only provide detailed cloud properties (e.g. liquid/ice water path, low/mid/high cloud amount) in ‘AMIP’ experiment over approximately 120 ‘cfSites’, which are not enough for the assessment of their radiative impacts at global scales.

We noted in the updated manuscript that future investigation on these issues are possible following the new approach developed in the revised paper, once the corresponding variables become available for analysis from the modeling groups.

Figure 5. Diurnal cycle of cloud radiative effects (CRE) in AMIP experiment over the ocean at ‘cfSites’ from climate models. After Webb et al. (2015).

The missing of the analysis of GCM simulated CRE make the readers hard to imagine the actual diurnal cycle of CRE in the models. It also makes it hard to connect the GCMs with the simple radiative balance model. Why should one trust the extremely simple radiative balance model if it is not demonstrated to be relevant to the comprehensive GCMs' results?

We now directly use the TOA radiative fluxes and CRE to estimate the DCC effects as explained in detail in Figure 4 and the responding text (see especially the new Methods section). By using the definition of CER, the TOA reference radiation is quantified as the clear-sky and cloud-sky radiation weighted by the cloud coverage. In this way, the Earth's radiation budget can be analyzed by varying the amplitude and phase of the cloud coverage. The analysis from this new approach is more accurate at representing the radiative fluxes from GCMs. Note the global mean TOA radiation for ERA-20C is about -0.9 Wm^{-2} , which is not perfectly representing the reality because reanalysis is not particularly designed for conserving energy balance (Berrisford et al. 2009; Dee et al. 2011). Therefore, the relative values of TOA reference radiations are more informative and should be used for quantifying DCC biases.

3) Based on the authors' metrics, the CNRM-CM4 model outperform most other GCMs in simulating the diurnal cycle of clouds. This provides a very good opportunity for the authors to investigate the implication and consequence of the diurnal cycle of clouds on climate projections since the contrast between good and bad diurnal cycle models would help to identify key roles of the diurnal cycle of clouds in climate simulation and projections. I hope the authors can pursue this.

We thank the reviewer for this constructive comment. In the revised manuscript, we explicitly showed the DCC of CNRM-CM5 (i.e. Fig.1 in the main text as reported here in Figure 6).

More importantly, we explored and compared the DCC feedbacks in CNRM-CM5. The results from CNRM-CM5 are consistent with other climate models, showing that the DCC does not have significant variations under future climate conditions so that the corresponding radiative impacts are much smaller than those from cloud cycle errors. This provides a good opportunity for us to justify that model tuning could still remain largely effective to provide reasonable climate projections. We thank the reviewer for this suggestion.

Figure 6. DCC climatology and empirical distribution of its indexes from ISCCP satellite records, ERA-20C reanalysis data, GCMs.

4) The GCM clouds are typically tuned to produce the observed global TOA radiative fluxes for both total and individual (LW and SW) components. This tuning is typically done in AMIP simulations forced by observed SSTs. Since the global surface temperature is largely

constrained by the specified SSTs, it is not really involved in calibrating the GCM clouds.

We agree that SST is constrained in AMIP experiment and therefore does not show the model tuning effects. In the revised manuscript, we selected ERA-20C as the **baseline** model and used its CRE to define TOA radiation for other climate models. We split the DCC into the mean and fluctuation part, $f(t) = \mu + f_{\text{DCC}}(t)$; see Methods section. We then use μ from ERA-20C and f_{DCC} from each climate model to re-calculate the TOA radiation, which is referred to as ‘**reference**’ radiation in the manuscript. Such a re-calculated variable involves the f_{DCC} effects and is more appropriate for the inter-model comparison. We did not explain this point very well in the original manuscript, but now have clarified how this ‘reference’ radiation is re-calculated and explained its usefulness for evaluating model tuning.

Moreover, there are many different ways GCM clouds can be tuned to produce the correct total flux. Liquid water path is only one of them. Thus, the authors' conclusion of overestimation of LWP is speculative.

We agree. Indeed, multiple climate variables may have errors and need to be tuned; multiple parameters can be adjusted to balance the Earth’s energy budget. This makes it hard to disentangle the precise error-adjustment mapping. In the updated manuscript, we have removed the speculative conclusion of overestimation of LWP.

We only commented on the potential factors that may compensate the impacts of cloud cycle errors. As suggested by the reviewer, we shifted our focus to analyze the implication of DCC biases to the climate projections. We found that DCC biases do not seem to invalidate future climate projection, but may induce overestimation of cloud-feedback strength and distort the patterns of land-ocean-atmosphere interaction.

It is also not very useful because it is well known about the compensation errors in tuning clouds. The question is what is the real consequence of the compensation errors. Does it matter for climate projections? Why or Why not?

To provide a more useful conclusion, we analyzed the implications of these DCC errors to the future climate projection.

We first analyzed the DCC radiative impacts under future climate conditions. As shown in Figure 3 and explained in the corresponding text, the DCC has much smaller radiative impacts in the climate projection than in the current climate conditions. Especially the CNRM-CM5 model, which has more accurate DCC simulations shows that the DCC plays a limited role in controlling energy budget in future climates. This indicates that model tuning could still remain largely effective to provide reasonable projections.

We also discussed the potential indirect impacts of DCC errors, including exaggerating cloud feedbacks and disturbing land-ocean-atmosphere interactions (see more details in the reply to the reviewer’s first major question).

We have added these analysis in the revised manuscript.

Minor comments:

There are a number of locations in the paper that miss proper definitions, explanations or references. Below are a few examples:

Line 37: What are exactly the cloud coverage variable used from the models? Do you distinguish between low, middle, and high cloud?

We used total cloud coverage without distinguishing low/middle/high clouds. Actually, we use the CRE from the reanalysis data imply that we implicitly consider all the cloud properties but do not specifically focus on their diurnal cycle errors. Although cloud structure has great impacts on Earth's longwave radiation, its diurnal cycle errors may have negligible longwave radiative impacts because DCC controls the energy budget primarily through the shortwave radiation (see Figure 5 and the corresponding text above).

We only focus on total cloud coverage is also due to the fact the model outputs do not make available other cloud properties (except AMIP experiment at limited 120 sites). We now have one section explicitly discussing the role diurnal cycle of other cloud properties and we emphasize that the new approach developed here may easily be generalized to other cloud variables, once they become available.

Do you use products from satellite simulator?

We did not use products from satellite simulator. Products from satellite simulator at sub-daily timescale usually are only from few climate models or experiments. For example, total cloud coverage from ISCCP simulator (cltiscpp) at 3-hour timescale are only available for TAMIP experiment in 2008 or 2009 from three GCM outputs. These data sets are not long enough to calculate the DCC climatology. In the revised manuscript, we have now clarified which data sets were used for the analysis.

Line 40-42: Please provide evidence or reference that the use of centroid defined in this paper compares well with other methods in determining the phase of diurnal cycle.

In the revised manuscript, we showed that the global maps of the centroid are very similar to maps of phase of the first harmonic as reported below in Figure 7. This figure is also included in the supplementary material of the manuscript.

Figure 7. DCC indexes and amplitude/phase of the first harmonic in winter (DJF) from CNRM-CM5 model.

Line 51-52: Why CNRM-CM4 relies on ECMWF IFS?

The atmospheric model ARPEGE-Climat used in CNRM-CM5 is derived from the ARPEGE/IFS jointly developed by Meteo-France and ECMWF (Volodire et al. 2013). We have updated the text and explicitly explained the relationship between CNRM-CM5 and IFS.

Line 104-106: This is what I hope the authors to focus on. I am disappointed that the authors did not pursue this further.

We now analyze the DCC in response to the climate change to assess its potential impacts to the climate projection. As shown in Figure 3 and explained in the corresponding text, DCC does not systematically shift so that its radiative impacts on the future climate change are limited. Therefore, model tuning made for current climate conditions still remains largely effective for the future climate projections.

Line 108-109 and line 313-348: The section of description of the radiative balance model lacks of explanation, references, and validation for the formulations, parameter setting, and model performance. It is not clear to what extent it is relevant for understanding the GCM simulated effect of diurnal cycle of clouds since no GCM results are compared.

We now directly use CRE (Cess et al. 1989, 1990, 1996; Ramanathan et al. 1989) to analyze the radiative effects of DCC. The TOA radiation is expressed as the clear-sky and cloudy-sky radiation weighted by the total cloud coverage. In this way, Earth's radiation budget can be analyzed by varying the amplitude and phase of the cloud coverage. (see details in Figure 4 and the response above).

Line 149-152: GCM clouds are typically tuned to produce correct global

TOA fluxes in simulations forced by observed SST. Moreover, there are many ways to tune the clouds, LWP is only one of them.

Indeed, multiple climate variables may have errors and need to be tuned; multiple parameters can be used to adjust the errors. As already explained previously in the response letter, we have acknowledged the typical tuning approach in climate models and discussed the involvement of other climate variables/parameters (see text for details).

Line 156-157: The correlation is not very good. Even it is good, it may be due to different reasons (e.g., cloud phase response to surface temperature).

We have removed the correlation analysis and discussed other ways of model tuning.

Line 159-161: Why not directly examine the diurnal cycle of TOA radiative flux or cloud radiative effects?

We now directly use the diurnal cycle of TOA fluxes and CRE to analyze the DCC radiative impacts (see details in Figure 4 and the response above).

Line 174-184: The paper does not have a strong conclusion.

To pursue a stronger conclusion, we now show the implication of DCC errors to the climate projections. We thank the reviewer for his/her constructive comments/suggestions, which have spurred us to analyze the DCC in response to the climate change. The reviewer's hint to pay more attention to CNRM-CM5 model helped us to identify the potential roles of DCC in climate projections. Thank you.

Reviewer #4 (Remarks to the Author):

This manuscript documents biases in the diurnal cycle of clouds in the CMIP5 model ensemble. The authors first claim that while the mean cloud cover in most models is fairly well captured, the amplitude and the phase of the diurnal cycle exhibit important biases. Over land, the amplitude is too small and the cloud cover peaks in the morning rather than in the afternoon. The oceanic diurnal cycle is better captured, but cloud cover still peaks too early in the night. The second part of the manuscript tries to explain the importance of these biases from a radiation point of view, by applying a minimalist radiative transfer model. This analysis reveals that the biases in the cloud diurnal cycle should lead to an excess radiation at the surface and in turn a very significant warm bias in all models. However, models tend to compensate for the lack of cloud by having too large in-cloud water contents, so that the average global radiative impact of clouds in most models is still fairly well captured.

We thank the reviewer for his/her comprehensive summary of the work. We have used these comments to improve the manuscript as addressed below.

The study of the diurnal cycle of clouds is important and to my knowledge this is the first study tackling this issue on a global scale.

We thank the reviewer for the positive comments and encouragement.

The method of decoupling the mean, amplitude and phase of the diurnal cycle seems sound, but I do wonder why the typical noon-peak in cloudiness over land in climate models is absent in their extended data figure 2 in most models. I wonder if the authors could comment on this.

We used the same codes to calculate the DCC phase for all climate models. As suggested by the reviewer later, the absence of non-peak in cloudiness might be accounted for by the parameterization of convection. Nonetheless, this feature has been well captured by the CNRM-CM5 model. We now explicitly display the results from CNRM-CM5 in Fig.1 of the main text and also reported here in Figure 8.

Figure 8. (1st column) DCC climatology in Guangde, China and (2nd-4th column) the empirical distribution of its indexes from ISCCP satellite records, ERA-20C reanalysis data, GCMs.

Furthermore, many studies on the diurnal cycle of clouds and precipitation in specific regions have been done before and as such, this study is not presenting an entirely surprising or novel result. Studies over the past 15 years on the diurnal cycle of clouds and precipitation in climate models include: Yang and Slingo (2001), Clark et al. (2007), Pfeifroth et al. (2012), Langhans et al. (2013), Walther et al. (2013) or Gustafson et al. (2015). These studies recognise for various regions on the

globe that the cloud diurnal cycle is poorly captured in climate models, but that this greatly improves when these models are run at convection-permitting scales (e.g. Clark et al. 2007, Langhans et al. 2013, Brisson et al 2016), suggesting the parameterization of convection is the main culprit for this deficiency.

We thank the reviewer for pointing out these relevant references. We have included them in the revised manuscript to provide a better introduction to the prior studies of the diurnal cycle of clouds. We also cited Clark et al. (2007), Langhans et al (2013), and Brisson et al (2016) to explain that parameterization of convection may be one of the important reason responsible for the missing noon-peak cloudiness over the land.

Moreover, the claim by the authors that the lack of cloudiness in the daytime is compensated by too large in-cloud liquid water content ties in with the well-documented “too few, too bright” problem that has been extensively studied as well, even in the context of the CMIP5 models (e.g. Williams and Webb 2009, Nam et al. 2012, Tsushima et al. 2015).

We appreciate the reviewer’s help to support our conclusions. We have revised the manuscript and explained that DCC biases may tie to the “too few, too bright” problem.

While I think there is certainly merit in the first part of their analysis, I’m not convinced by the minimalistic radiative transfer model approach to show the possible impact of the cloud biases on the radiation balance.

To avoid potential doubts regarding the minimalist model and its limitations to simulate the impacts of all types of cloud properties on earth energy budget, in the updated manuscript, we directly use the CRE and TOA radiative fluxes from climate models to analyze the radiative impacts of cloud cycle biases. We start from CRE, which are conventionally defined as the difference of TOA all-sky (R) and clear-sky (R_{clr}) net radiative flux (Cess et al. 1989, 1990, 1996),

$$\text{CRE} = R - R_{\text{clr}}. \quad (9)$$

This quantity can also be expressed in terms of cloud coverage as in Eq. 4 of Ramanathan et al. (1989),

$$\text{CRE} = f(R_{\text{cld}} - R_{\text{clr}}), \quad (10)$$

where R_{cld} is the cloudy-sky radiative flux. Combining (9) and (10), R_{cld} can thus be expressed as

$$R_{\text{cld}} = \frac{1}{f}[R - (1 - f)R_{\text{clr}}]. \quad (11)$$

Since all-sky, clear-sky radiative fluxes, and total cloud coverage are often provided in GCM outputs, R_{cld} can be calculated directly from GCM outputs using (11).

With known values of R_{cld} , it is now possible to recalculate the TOA radiative flux as a function of cloud coverage and its properties, by solving (11) for R ,

$$R = fR_{\text{cld}} + (1 - f)R_{\text{clr}}, \quad (12)$$

Specifically, with (12) we can use the R_{clr} and R_{cld} provided by GCMs or other data sources to analyze the impacts of diurnal variations of cloud coverage on TOA radiation. This radiation used for analyzing DCC errors is referred to as TOA reference radiation.

Figure 4 shows both the diurnal cycle of CRE climatology and the corresponding TOA reference radiation as functions of centroid and standard deviation of cloud cycle.

Figure 9. CRE and TOA reference radiation. Diurnal cycle of CRE climatology over land (a) and ocean (b) from ERA-20C reanalysis; ‘heatmap’ of TOA reference radiation over the land (c) and ocean (d) as functions of coefficient of variation and centroid of cloud diurnal cycles. The crosses specify the 25th, 50th, and 75th percentiles of the c and c_v from GCMs (open markers), ISCCP records (filled hexagram), and ERA-20C reanalysis data (filled circle). Note the 10th, 50th, and 90th percentiles of c and c_v over the ocean were shown in the original manuscript but has been changed to the consistent 25th, 50th, and 75th in the revised manuscript.

The authors come up with a simple 1D radiative transfer model and plug in the globally averaged values for cloud albedo and water path in their calculations.

We have now assessed TOA radiative fluxes from reanalysis at each grid point in each season to evaluate the radiative impacts of cloud cycle. The CRE and TOA reference radiation in Figure 9 are show the (approximate) radiation impacts as a function of phase and amplitude of DCC. A complementary assessment (reported here in Figure 10) accounts for the detailed spatial and seasonal variations. We also note in the revised manuscript that, since the data assimilation is not designed to conserve the energy balance (Berrisford et al. 2009; Dee et al. 2011) and the long-term global mean TOA radiation from ERA-20C probably does not perfectly represent the reality. The relative values between those from GCMs and the standard ERA-20C are more informative and should be used to quantify the DCC radiative impacts.

Figure 10. TOA reference radiation averaged in each grid point and each season over the land, ocean, and the whole Earth with mean cloud coverage from ERA-20C reanalysis and the diurnal fluctuation climatology from each data source during 1986-2005. The dash lines show the TOA reference radiation from ERA-20C reanalysis.

Furthermore, they apply this model on a diurnal cycle of cloud cover including their observed and simulated values of the mean, amplitude and phase. In their supplementary figures, they show large variations in these three diurnal cycle statistics, not necessarily with a good correlation between the biases in these three parameters. Given that radiative transfer is a highly non-linear process, I feel that this 1D radiative transfer model bears very little resemblance to what the real impact of cloud biases in the CMIP5 models on their global radiation balance would be. Indeed, it matters a lot where biases in amplitude and phase are overlapped or where the biases in water path and amplitude are overlapped.

To avoid potential limitations of 1D radiative transfer model, we have removed this analysis in the updated manuscript. Instead, to evaluate the impacts of DCC errors we now directly use the CRE and TOA radiative fluxes from the state-of-the-art ERA-20C reanalysis. We have showed the global mean CRE and TOA reference radiation in Figure 9 to demonstrate the DCC radiative impacts on Earth's energy budget. Moreover, we also used the CRE at each grid point in each season to provide complementary assessment TOA reference radiation as in Figure 10.

Furthermore, an important simplification is the fact that the authors do not discriminate between high clouds and low clouds in their analysis. High clouds are known to have a predominant warming effect, while liquid clouds have a predominant cooling effect on earth's climate (Ramanathan et al. 1989).

We agree with the reviewer that cloud structure has significant impacts on the Earth's energy balance. First of all, in the updated manuscript, we now use the CRE and TOA radiation from the state-of-the-art ERA-20C reanalysis, whose cloud coverage data are impacted by all important cloud properties (e.g. cloud heights, water path, liquid/ice water content).

Secondly, the fact that we limit our analysis to the cloud coverage $f(t)$ is due to data availability from GCMs. However, following the reviewer's comments, we have extended our discussion to the effects of other cloud properties. For example, are there diurnal cycle **errors** of other cloud properties? Are the radiative impacts of these errors important? These questions now have been discussed in the last section of the revised manuscript and are reported below:

As the reviewer pointed out, the cloud structure controls the radiation budget primarily through the longwave components. In one independent study from Webb et al. (2015), the authors have shown the longwave CRE has much smaller diurnal cycle than its shortwave counterpart (see Figure 11 below). Our baseline CRE from ERA-20C reanalysis also shows similar results (see Figure 9 a and

b). These results explain that diurnal cycles of clouds adjust the energy balance primarily through its modulation on shortwave radiation. Therefore, even if there are diurnal cycle errors of cloud heights among climate models, these errors likely have much smaller longwave radiative impacts. Note that we only suggest that the **diurnal cycle errors of cloud height** are less important, while obviously the **daily mean cloud height errors** could still have significant radiative impacts.

We also acknowledged in the manuscript that the diurnal cycle of cloud heights can change the cloud-top temperature and consequently induces the change of liquid/ice water content and cloud albedo. In this way, diurnal cycle errors of cloud heights also influence the shortwave radiation and need to be analyzed.

We now have a specific section in the revised manuscript discussing the impacts of diurnal cycle of in-cloud properties (e.g. cloud structure, liquid/ice water content, and cloud microphysics). We also stress that these quantities can be analyzed using the new framework developed in the revised manuscript for $f(t)$, once the necessary data become available from the modeling groups.

Figure 11. Diurnal cycle of cloud radiative effects (CRE) in AMIP experiment from climate models. After Webb et al. (2015).

By simply assuming a globally averaged cloud albedo and water content, this important distinction between ice and liquid clouds is ignored.

Aside from using globally averaged CRE to demonstrate the DCC radiative effects as functions of DCC phase and amplitude in Figure 9, we also assess the TOA reference radiation at each grid point in each season and then averaged over the region under consideration as shown in Figure 10. Therefore, the important distinctions between the ice and liquid clouds at different local points in different seasons are considered in the analysis.

Hence, it is very likely that the suggested very large impact of the CMIP5 bias in the cloud diurnal cycle on the radiation balance is incorrect. This is also reflected in the fact that the simple radiative transfer model only reaches a mean global temperature of 285.5 K (Figure 4), which is 2-3 K colder

than the observed global mean temperature (or even lower, given that the authors omit the poles from their analysis).

We agree that the simple model may have limitations and therefore it was removed from the manuscript. As said before, in the updated manuscript, we have adopted a new approach which directly uses CRE from the ERA-20C reanalysis to evaluate TOA radiations. Note that reanalysis is not particularly designed to conserve the energy balance (Berrisford et al. 2009; Dee et al. 2011) and ERA-20C introduces some artificial energy source (i.e. the long-term mean global net radiation is -0.9 Wm^{-2} , see Figure 10). The relative values of reference radiation are more informative and should be used for quantifying DCC radiative impacts.

Last, I have a minor comment about Figure 2. I'm not sure whether these contingency tables are the best way to convey the message here. Wouldn't it be better to just show bar plots of the mean observed (ISCCP and ERA) and each of the CMIP modelled mean cloud cover, amplitude and phase? Or alternatively, just bar plots of the RMSD between the observations and each of the CMIP models. I'm not sure what we learn from the RMSD between each of the different models.

We followed the second suggestion and used the bar plots to show the RMSD between ISCCP observations and each CIMP models (reported below in Figure 12). Using mean values may convey a less compelling message. For example, the centroid in most climate models looks like uniform distribution (see Figure 8), which results in mean value close to noon (12 hours). The centroid in ISCCP/ERA are clearly unimodal distributions with mean also close to noon (12 hours). Therefore, mean values of centroid are not efficient at distinguishing the difference between simulations and observations.

Figure 12. Root-mean-square deviation (RMSD) of DCC indexes between ISCCP satellite observations and model outputs.

Based on the fact that the first part of this paper is relevant and well executed, but not entirely novel, and that the second part of the analysis seems to be a too simple approach to support their fairly far-reaching conclusion, I would not recommend this paper to be accepted in its current form. I think a far more advanced approach is required to show the impact of current cloud biases in climate models on the radiation balance.

We thank the reviewer for the constructive and encouraging comments. We have sought a more advanced approach using TOA radiative fluxes and CRE directly from model outputs to assess the radiative impacts of DCC errors. These additions have provided much more strength to our conclusions, as detailed in the responses above and in the thoroughly revised manuscript. We hope that this approach will provide more accurate evaluation and stress the importance of convections parameterizations.

References suggested by the reviewer:

Brisson, E., Van Weverberg, K., Demuzere, M. et al. (2016). How well can a convection-permitting climate model reproduce decadal statistics of precipitation, temperature and cloud characteristics? Clim Dyn 47: 3043.

Clark, A. J., Gallus, W., Chen, T.-C. (2007). Comparison of the diurnal precipitation cycle in convection-resolving and non-convection-resolving mesoscale models. Monthly Weather Review. 135, 3456-3473.

Gustafson Jr., W.I., Ma, P.-L., Singh, B. (2015). Precipitation characteristics of CAM5 physics at mesoscale resolution during MC3E and the impact of convective timescale choice. Journal of Advances in Modeling Earth Systems 6, 1271-1287.

Langhans W, Schmidli J, Fuhrer O, Bieri S, Schär C (2013). Long-term simulations of thermally-driven flows and orographic convection at convection-parameterizing and cloud-resolving resolutions. J Appl Meteorol Climatol 1:130117155938002 (1984)

Nam C, Bony S, Dufresne J, Chepfer H (2012). The “too few, too bright” tropical low-cloud problem in CMIP5 models. Geophys Res Lett. doi:10.1029/2012GL053421

Pfeifroth U, Hollmann R, Ahrens B (2012). Cloud cover diurnal cycles in satellite data and regional climate model simulations. Meteorol Zeitschrift 21(6): 511-560

Ramanathan, V., R. D. Cess, E. F. Harrison, P. Minnis, B. R. Barkstrom, E. Ahmad and D. Hartmann (1989). Cloud-Radiative Forcing and Climate: Results from the Earth Radiation Budget Experiment. Science New Series, 243, 57-63

Tsushima Y., Ringer, A.M., Tsuyoshi, K., Kawai, H., Roerhig, R., Cole, J., Watanabe, M., Yokohata, T., Bodas-Saledo, A., Williams, K., Webb, M. (2015). Robustness, uncertainties, and emergent constraints in the radiative responses of stratocumulus cloud regimes to future warming, Clim Dyn, DOI 10.1007/s00382-015-2750-7

Walther, A., Jeong, J.-H., Nikulin, G., Jones, C., Chen, D. (2013). Evaluation of the warm season diurnal cycle of precipitation over Sweden simulated by the Rossby Centre regional climate model RCA3. *Atmospheric Research*. 119, 131-139.

Williams K, Webb M (2009). A quantitative performance assessment of cloud regimes in climate models. *Clim Dyn* 33(1):141157.

Yang, G.Y., Slingo, J. (2001). The diurnal cycle in the Tropics. *Monthly Weather Review* 129, 784-801.

We thank the reviewer for providing these useful references. We have cited them to provide a better literature review of DCC and explain that parameterization of convection may be responsible for DCC biases in simulations.

Reference

Berrisford, P., Dee, D., Fielding, M., Fuentes, P., Kallberg, S., Kobayashi, and S. Uppala, 2009: The ERA-interim archive. *ERA Rep. Ser.*, 1–16.

Bony, S., M. Webb, C. S. Bretherton, S. A. Klein, P. Siebesma, G. Tselioudis, and M. Zhang, 2011: CFMIP: Towards a better evaluation and understanding of clouds and cloud feedbacks in CMIP5 models. *Clivar Exch.*, **56**, 20–22.

Brient, F., and S. Bony, 2012: How may low-cloud radiative properties simulated in the current climate influence low-cloud feedbacks under global warming? *Geophys. Res. Lett.*, **39**, L20807, doi:10.1029/2012GL053265.

Cairns, B., 1995: Diurnal variations of cloud from ISCCP data. *Atmospheric Res.*, **37**, 133–146.

Cess, R. D., and Coauthors, 1989: Interpretation of Cloud-Climate Feedback as Produced by 14 Atmospheric General Circulation Models. *Science*, **245**, 513–516, doi:10.1126/science.245.4917.513.

——, and Coauthors, 1990: Intercomparison and interpretation of climate feedback processes in 19 atmospheric general circulation models. *J. Geophys. Res. Atmospheres*, **95**, 16601–16615, doi:10.1029/JD095iD10p16601.

——, and Coauthors, 1996: Cloud feedback in atmospheric general circulation models: An update. *J. Geophys. Res. Atmospheres*, **101**, 12791–12794, doi:10.1029/96JD00822.

Dee, D. P., and Coauthors, 2011: The ERA-Interim reanalysis: configuration and performance of the data assimilation system. *Q. J. R. Meteorol. Soc.*, **137**, 553–597, doi:10.1002/qj.828.

Karlsson, J., G. Svensson, and H. Rodhe, 2008: Cloud radiative forcing of subtropical low level clouds in global models. *Clim. Dyn.*, **30**, 779–788, doi:10.1007/s00382-007-0322-1.

Nam, C., S. Bony, J.-L. Dufresne, and H. Chepfer, 2012: The “too few, too bright” tropical low-cloud problem in CMIP5 models. *Geophys. Res. Lett.*, **39**, L21801, doi:10.1029/2012GL053421.

- Ramanathan, V., R. D. Cess, E. F. Harrison, P. Minnis, B. R. Barkstrom, E. Ahmad, and D. Hartmann, 1989: Cloud-radiative forcing and climate: results from the Earth radiation budget experiment. *Science*, **243**, 57–63, doi:10.1126/science.243.4887.57.
- Rozendaal, M. A., C. B. Leovy, and S. A. Klein, 1995: An Observational Study of Diurnal-Variations of Marine Stratiform Cloud. *J. Clim.*, **8**, 1795–1809, doi:10.1175/1520-0442(1995)008<1795:Aosodv>2.0.Co;2.
- Taylor, K. E., R. J. Stouffer, and G. A. Meehl, 2011: An Overview of CMIP5 and the Experiment Design. *Bull. Am. Meteorol. Soc.*, **93**, 485–498, doi:10.1175/BAMS-D-11-00094.1.
- Tsushima, Y., and Coauthors, 2016: Robustness, uncertainties, and emergent constraints in the radiative responses of stratocumulus cloud regimes to future warming. *Clim. Dyn.*, **46**, 3025–3039, doi:10.1007/s00382-015-2750-7.
- Voltaire, A., and Coauthors, 2013: The CNRM-CM5.1 global climate model: description and basic evaluation. *Clim. Dyn.*, **40**, 2091–2121, doi:10.1007/s00382-011-1259-y.
- Webb, M. J., and Coauthors, 2015: The diurnal cycle of marine cloud feedback in climate models. *Clim. Dyn.*, **44**, 1419–1436, doi:10.1007/s00382-014-2234-1.
- Williams, K. D., and M. J. Webb, 2009: A quantitative performance assessment of cloud regimes in climate models. *Clim. Dyn.*, **33**, 141–157, doi:10.1007/s00382-008-0443-1.
- Wylie, D. P., and H. M. Woolf, 2002: The Diurnal Cycle of Upper-Tropospheric Clouds Measured by GOES-VAS and the ISCCP. *Mon. Weather Rev.*, **130**, 171–179, doi:10.1175/1520-0493(2002)130<0171:TDCOUT>2.0.CO;2.

Reviewer #1 (Remarks to the Author):

This manuscript is much improved and has addressed the issues raised in the previous review. The use of the CREs helps avoid the issue of dealing with all of the various cloud properties affected by the diurnal cycle. It is ok for publication after correcting some minor grammatical errors.

Line 10: "terrestrial radiation", not "terrestrial one"

44: "implication" should be plural to match the verb

68: "model" should be plural to match the verb

106: "is" should be "are" to match the subject

171 and elsewhere: "radiations" is not a very precise term. "radiative fluxes" or "irradiances" would be more appropriate.

179: "late" should be "later"

The reference for #6 should not be "Part I", but
Part III: November 1978 radiative parameters. J. Clim. Appl. Meteorol., 23, 1032-1052.

Reviewer #2 (Remarks to the Author):

I am glad to see that the paper has been improved, especially with the new focus on the effect of cloud cycle biases in climate models. However, I don't think the authors have made enough of an effort to connect their analysis to the actual model results. Their simple model uses only some statistics of the GCM simulated total cloud cover, with the rest of the statistics (clear-sky, total-sky and cloudy-sky CRE) being obtained from ERA-40. Moreover, they do not try to verify the extent to which their simplifications can reproduce features of the actual GCM simulated CREs including their inter-model variability and change with global warming. For example, to what extent do the results in Fig. 4 and Fig. 5 correlate with the actual GCM results? What are the effects of compensating tuning in other cloud properties? What is the role of vertical structures of clouds given low and high clouds have opposite effects on radiation? The ignorance of detailed analysis of actual GCM simulated clouds and CREs make it hard to connect the simple analysis to actual GCM results. In addition, why do the authors use ERA-40 for CRE calculations instead of using the actual observations (e.g., CERES)? If the authors choose ERA-40, they would need to make a comparison of ERA-40 to CERES observations to establish the validity of ERA-40 for this study.

Reviewer #3 (Remarks to the Author):

The authors have made major changes that I think have made the manuscript a lot better. They removed the very simple radiative transfer model and replaced it by a top of atmosphere analysis to separately assess the impact of the mean radiation biases and the diurnal cycle of radiation. This approach is much more sensible and better supports the conclusions of the study. I think this is a nice and original study, and I therefore recommend to accept the manuscript for publication, pending a minor comment.

On line 87-88, the authors state that the CNRM-CM5 reproduces the DCC much better over land, which the authors attribute to the more sophisticated model. Could the authors be a bit more elaborate on what aspect exactly makes this model so much more sophisticated? Is the convection scheme in this model more advanced than for instance in HadGEM? I realize that the scope of this work is not to give conclusive answers on the reason why some models perform better than others, but it would be good to provide little bit of an hypothesis about this.

Further, there are a few typos that I list below:

L 61: change 'derived' to 'derive'.

L78: change 'convections' to 'convection'

L180: suggest rephrasing to 'Other GCMs instead have larger reference radiation that is...'

Each reviewer comment (italicized) is followed by a response.

Reviewer #1:

Reviewer #1 (Remarks to the Author):

This manuscript is much improved and has addressed the issues raised in the previous review. The use of the CREs helps avoid the issue of dealing with all of the various cloud properties affected by the diurnal cycle. It is ok for publication after correcting some minor grammatical errors.

We thank the reviewer for the encouragement.

Line 10: "terrestrial radiation", not "terrestrial one"

Corrected.

44: "implication" should be plural to match the verb

Corrected.

68: "model" should be plural to match the verb

Corrected.

106: "is" should be "are" to match the subject

Corrected.

171 and elsewhere: "radiations" is not a very precise term. "radiative fluxes" or "irradiances" would be more appropriate.

Corrected.

179: "late" should be "later"

Corrected.

*The reference for #6 should not be "Part I", but
Part III: November 1978 radiative parameters. J. Clim. Appl. Meteorol., 23, 1032-1052.*

Corrected.

We thank the reviewer again for the comments and encouragement received throughout the review process.

Reviewer #2 (Remarks to the Author):

I am glad to see that the paper has been improved, especially with the new focus on the effect of cloud cycle biases in climate models.

We are glad the reviewer appreciated our effort in improving our paper.

However, I don't think the authors have made enough of an effort to connect their analysis to the actual model results. Their simple model uses only some statistics of the GCM simulated total cloud cover, with the rest of the statistics (clear-sky, total-sky and cloudy-sky CRE) being obtained from ERA-40.

We have done our best to address this point. Toward this goal, we have extended our analysis, using the same method, to the 3-hourly global CRE from AMIP experiments in CanAM4 model (as far as we know, this is only this climate model that provides global subdaily CRE outputs for AMIP experiments; other models only provide the outputs for 120 'cfSites').

Figure 1 reported below compares the TOA reference radiations with the two different CRE data sources. As can be seen, the section of CRE data sources influences the absolute values of TOA reference radiation but has much less impacts on the inter-model patterns, which have been used to evaluate the DCC radiative impacts. Figure 2 compares the TOA reference radiations in response to the climate change. It also suggests that the choice of data sources for CRE has limited impact on the assessment of DCC impacts.

It is also interesting to note how the climate model CanAM4 may have been constrained to conserve energy so that its long-term mean global radiative flux of 0.16 Wm^{-2} is closer to the ocean heat uptake (Hansen et al. 2011). This notwithstanding, CanAM4, like the other climate models already analyzed in the manuscript, tends to simulate the clouds that peak too earlier over the land, so that its TOA radiation may have been influenced by model tuning. For this reason, CNRM-CM5 with its more reliable simulation of DCC now tends to show some biases of TOA reference radiations (see bottom panel of Figure 1). Comparing with CanAM4, CNRM-CM5 reflects more solar radiation at noon over the land thus resulting in lower reference radiation.

We updated the manuscript to explain the additional analyses and compared the results from ERA-20C reanalysis and from CanAM4 climate model. The similar patterns of these results were used to explain that CRE sources may have limited impacts on the assessment of DCC.

Figure 1 TOA reference radiation with CRE from ERA-20C reanalysis (top panel) and from CanAM4 AMIP experiment (bottom panel). The red dash and blue dash lines are the TOA radiation from ERA-20C and CanAM4, respectively.

Figure 2 DCC responses to climate change in terms of change of TOA reference radiation with CRE obtained from ERA-20C reanalysis (top panel) and from CanAM4 AMIP experiment (bottom panel).

Moreover, they do not try to verify the extent to which their simplifications can reproduce features of the actual GCM simulated CREs including their inter-model variability and change with global warming.

We agree with the reviewer (if we correctly understand this comment) that it would be crucial to carefully compare the proposed procedure with simulated CREs. To fully address this issue, however, would imply re-running the radiation codes in each climate model to get the necessary outputs at the desired resolution and analyze the inter-model variability of TOA reference radiation. This would require access to CGM and collaboration with the climate modelling groups over the world. This is obviously not in the authors' (or anyone for that matter) reach, and certainly beyond the scope of the paper. Indeed, we hope that our analyses will stimulate such type of endeavor in joint programs of GCM comparisons.

More importantly for the specific revision here, the new results from CanAM4 and ERA-20C in Figure 1 and Figure 2, suggest that the CREs from different climate models have limited impacts on inter-model patterns of TOA reference radiations, thus confirming that the selection of CRE data sources is less important and the results from ERA-20C/CanAM4 are accurate enough for the assessment of DCC radiative impacts.

For example, to what extent do the results in Fig. 4 and Fig. 5 correlate with the actual GCM results? What are the effects of compensating tuning in other cloud properties?

Although model tuning will influence the absolute values of TOA reference radiations, it has limited impacts on their relative values (see Figure 1 and Figure 2). In fact, the inter-model patterns of TOA reference radiations are very similar, which are good enough for the purpose of evaluating the DCC radiative impacts. This also suggests that the results of Fig. 4 and Fig 5 in the manuscript should be strongly correlated with other GCM results.

We have pointed this out in the revised manuscript and added the new figures in the supplementary material.

What is the role of vertical structures of clouds given low and high clouds have opposite effects on radiation?

Low clouds tend to emit more longwave radiation thereby cooling the earth; high clouds tend to emit less longwave radiation with warming effects. Since we are directly using CRE from reanalysis data and climate model outputs, such opposite effects of cloud structures are fully considered in the evaluation of DCC radiative impacts.

The ignorance of detailed analysis of actual GCM simulated clouds and CREs make it hard to connect the simple analysis to actual GCM results.

We have now added the analysis of the CRE from the actual climate model CanAM4 to evaluate the DCC radiative impacts as shown in Figure 1 and Figure 2. As can be seen, the results are quite similar to these from ERA-20C reanalysis. This should corroborate our results and connect more

directly to actual GCM results. We have updated the manuscript to make this point and added the new figures in the supplementary material.

In addition, why do the authors use ERA-40 for CRE calculations instead of using the actual observations (e.g., CERES)? If the authors choose ERA-40, they would need to make a comparison of ERA-40 to CERES observations to establish the validity of ERA-40 for this study.

We thank the reviewer for the suggestion. We added a comparison of the CREs from ERA-20C with those from CERES and CanAM4 in Figure 3. As can be seen, CRE climatology from these data sources has similar diurnal cycles. CRE from both ERA-20C reanalysis and CanAM4 climate model are slightly lower than those from CERES, which have also been observed by Calisto et al. (2014). Such differences may be linked to the different methods of observing/modeling clear-sky radiations. While climate models can run their radiation codes to find clear-sky radiation even for regions with 100% cloud coverage, satellite observations cannot. This 100% cloud coverage regions are quite normal especially over the Inter Tropical Convergence Zone.

Nonetheless, given that the diurnal cycles of CREs from CERES, ERA-20C, and CanAM4 are quite similar, we would expect their inter-model patterns of TOA reference radiations used for assessing DCC effects are also very similar. We have revised the manuscript and added the new figure in the supplementary material.

We thank again the reviewer for his/her positive criticism and constructive comments. We hope that the improvements made to the manuscript suitable address his/her concerns.

Figure 3 Comparison of diurnal cycle of CRE climatology over the land and ocean from CERES, ERA-20C, and CanAM4.

Reference

Calisto, M., D. Folini, M. Wild, and L. Bengtsson, 2014: Cloud radiative forcing intercomparison between fully coupled CMIP5 models and CERES satellite data. *Ann Geophys*, **32**, 793–807, doi:10.5194/angeo-32-793-2014.

Hansen, J., M. Sato, P. Kharecha, and K. von Schuckmann, 2011: Earth's energy imbalance and implications. *Atmospheric Chem. Phys.*, **11**, 13421–13449.

Reviewer #3 (Remarks to the Author):

The authors have made major changes that I think have made the manuscript a lot better. They removed the very simple radiative transfer model and replaced it by a top of atmosphere analysis to separately assess the impact of the mean radiation biases and the diurnal cycle of radiation. This approach is much more sensible and better supports the conclusions of the study. I think this is a nice and original study, and I therefore recommend to accept the manuscript for publication, pending a minor comment.

We thank the reviewer for his/her constructive comments and encouragement.

On line 87-88, the authors state that the CNRM-CM5 reproduces the DCC much better over land, which the authors attribute to the more sophisticated model. Could the authors be a bit more elaborate on what aspect exactly makes this model so much more sophisticated? Is the convection scheme in this model more advanced than for instance in HadGEM? I realize that the scope of this work is not to give conclusive answers on the reason why some models perform better than others, but it would be good to provide little bit of a hypothesis about this.

We thank the reviewer for his/her suggestions. As the reviewer commented in the last round of the revision, DCC is poorly captured in climate models, but can be greatly improved when climate models are run at convection-permitting scales. This suggests parameterization of convection and modelling resolutions may be critical to the simulation of DCC. We therefore followed the reviewer's suggestion and made the hypothesis that CNRM-CM5 may have advanced convection schemes or finer resolutions.

Further, there are a few typos that I list below:

L 61: change 'derived' to 'derive'.

Corrected

L78: change 'convections' to 'convection'

Corrected

L180: suggest rephrasing to 'Other GCMs instead have larger reference radiation that is...'

Corrected. Thank you.

REVIEWERS' COMMENTS:

Reviewer #3 (Remarks to the Author):

It is now ok for publication.